# OrdShap: Feature Position Importance for Sequential Black-Box Models

**Davin Hill***
Northeastern University

**Brian L. Hill**†
Age Bold

**Aria Masoomi**
Northeastern University

**Vijay S. Nori**
Optum AI

**Robert E. Tillman**
Optum AI

**Jennifer Dy**
Northeastern University

## Abstract

Sequential deep learning models excel in domains with temporal or sequential dependencies, but their complexity necessitates post-hoc feature attribution methods for understanding their predictions. While existing techniques quantify feature importance, they inherently assume fixed feature ordering — conflating the effects of (1) feature values and (2) their positions within input sequences. To address this gap, we introduce *OrdShap*, a novel attribution method that disentangles these effects by quantifying how a model's predictions change in response to permuting feature position. We establish a game-theoretic connection between OrdShap and Sanchez-Bergantiños values, providing a theoretically grounded approach to position-sensitive attribution. Empirical results from health, natural language, and synthetic datasets highlight OrdShap's effectiveness in capturing feature value and feature position attributions, and provide deeper insight into model behavior.

## 1 Introduction

As complex and opaque deep learning models are increasingly used in high-stakes applications, it is important to understand the factors that contribute to their predictions. *Feature attribution* methods are a widely used approach that seeks to quantify the sensitivity of model predictions to changes in individual input features, thus generating scores that represent that feature's importance or relevance [5, 89]. In particular, *local* methods generate separate scores for each given input sample; this helps users to understand individual model predictions rather than relying solely on global attribution scores. Many recent feature attribution methods have adapted the Shapley Value framework [65] from the field of cooperative game theory [43, 75]. Shapley-based methods have demonstrated their utility in a wide range of domains [70, 28, 9] and satisfy a number of theoretical properties [43].

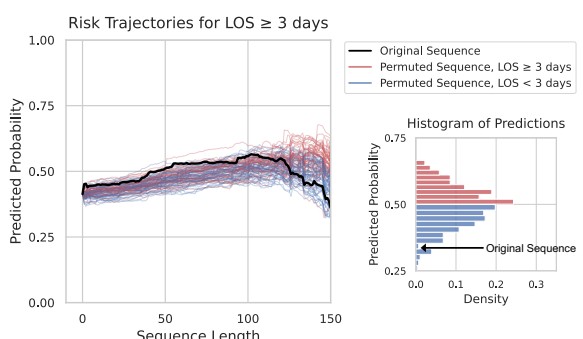

Figure 1: Predicting hospital Length-of-Stay (LOS) $\geq 3$ days for a patient using a sequence of medical tokens, representing tests and medications (App. A.2). Traditional feature attribution methods capture model sensitivity to token *values*. However, we observe that permuting token *order* has a significant effect on the predicted risk over time (**Left**) and final prediction (**Right**), even when token values are unchanged.

---

*Corresponding author. Email: hill.davi@northeastern.edu
†Work done while at Optum AI

39th Conference on Neural Information Processing Systems (NeurIPS 2025).

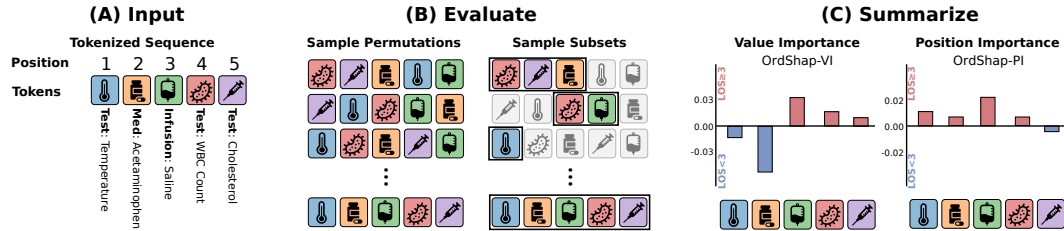

Figure 2: **Overview of OrdShap on a medical example**. **(A)** OrdShap takes a sequence of features (i.e. tokens) as input, then **(B)** evaluates the black-box model while sampling different token permutations (i.e. reordering tokens) and subsets. **(C)** We summarize attributions due to token *value* and token *position* using OrdShap-VI and OrdShap-PI, respectively. Positive OrdShap-VI indicates that the presence of the token in the sequence increases the probability of LOS$\geq 3$. Positive OrdShap-PI indicates that probability of LOS$\geq 3$ increases when the token appears later in the sequence; high magnitude indicates high model sensitivity to token position.

In this work we focus on sequential data, where the samples consist of a sequence of features, such as text, time-series, or genetic sequencing. Deep learning models such as Transformers [80] and Recurrent Neural Networks [62] have been shown to be highly effective on sequential data [83] and are able to capture the sequential dependency between data features. These models make predictions based not only on the specific feature values in each sample, but also the order in which they occur.

While existing feature attribution methods can be applied to sequential models, they inherently assume a fixed feature ordering. In particular, most approaches generally derive attributions by measuring the sensitivity of predictions to small changes in feature *values*, and otherwise disregard the effects of feature order [26]. However, permuting feature order, while leaving feature values unchanged, can have a significant impact on the model's prediction. For example, when predicting hospital length-of-stay from longitudinal Electronic Health Record (EHR) data[1] [59, 41, 55, 27], simply permuting the timestep of a test result can inflate the predicted risk even when the test result values are unchanged (Fig. 1), resulting in widely varying predictions. Since existing feature attribution methods assume a fixed feature ordering, they effectively conflate the effects of 1) the feature's value and 2) the feature's relative position within the sequence.

In this work we propose OrdShap, a local feature attribution approach that quantifies model sensitivity to both feature values and feature positions in sequential data (Fig. 2). To the best of our knowledge, OrdShap is the first method to disentangle these two effects. This disentanglement is particularly valuable in critical domains such as health, where the timing of medical events can be as important as their occurrence. For example, when analyzing patient trajectories, OrdShap can identify whether an elevated lab value is significant due to its magnitude or because it occurred when the patient was first admitted—a distinction that cannot be captured with existing methods.

**Main Contributions:**

- We propose OrdShap, a local feature attribution framework for sequential models. We further propose OrdShap-VI and OrdShap-PI, which quantify the Value Importance (VI) and Position Importance (PI) of each feature in a sequence, respectively.
- We establish a game-theoretical connection between OrdShap and the Sanchez-Bergantiños (SB) value [63], which satisfies several desirable axioms but has not been utilized for feature attribution.
- We propose two algorithms to efficiently approximate OrdShap.
- Empirical results from health, natural language, and synthetic datasets show that OrdShap is able to capture how feature ordering affects model prediction.

## 2   Related Works

**Feature Attribution Methods.** A variety of post-hoc feature attribution methods have been proposed [26, 88], many of which can be applied to sequential models. Existing methods have used feature masking [60, 43, 17], model gradients [4, 66, 76, 69], or self-attention weights [86, 1, 22, 13]. Lundberg and Lee [43] adapted Shapley Values [65] for feature attributions, which has been extended

---

[1]We provide additional background on EHR data in App. A.2.

to a number of Shapley-based approaches [75, 44, 21, 33]. While these attributions methods can be applied to sequential models, many assume feature independence [43, 39]. Frye et al. [23] and Wang et al. [82] investigate additional constraints to enforce sequential dependency. Second-order methods [44, 77, 46, 32, 79] can also be used to detect used to detect feature dependencies.

**Feature Attribution for Sequential Models.** Several methods have also been proposed specifically for sequential models [51, 89, 78], both on unstructured and structured datasets. TransSHAP [36] and MSP [72] calculate feature attributions for unstructured text using feature masking. TimeSHAP [6] adapts KernelSHAP [43] to Recurrent Neural Networks. Meng et al. [47] applies realistic perturbations to test samples using a generative model. TIME [71] calculates the change in model loss when permuting features between samples. While these methods are specific for sequential models, they do not explicitly estimate the importance of feature position or order. The most closely related work is PoSHAP [20], which is a global method that averages KernelSHAP [43] attributions over each position index for a given test set of samples. In contrast, the proposed method OrdShap is local attribution method that distentangles the effects of within-sample feature permutations.

## 3 Technical Preliminaries and Background

Let $X$ be a dataset, where each sample $x \in X$ is a sequence $(x_1, x_2, \ldots, x_d)$. We refer to elements $x_i \in x$ as *features* of $x$; depending on the application, these can represent (e.g.) tokens, sensor measurements, or word embeddings, and can be real or vector-valued. Given a sample $x \in X$, let $N = \{1, \ldots, d\}$ be the set of feature indices in $x$. Consider a trained prediction model $f$ that takes samples from $X$ as input. We refer to $f$ as the *black-box* model that we want to explain. We assume that $f$ has real-valued output; in the multi-class setting, we take the model output for a single selected class. Let $\mathfrak{S}_S$ be the symmetric group on $S \subseteq N$ with permutations $\sigma \in \mathfrak{S}_S$. We denote the bijective mapping corresponding to each $\sigma \in \mathfrak{S}_S$ using the same symbol, i.e. $\sigma(i)$ maps $i \in S$ to another element $j \in S$. For example, we can write a given $\sigma \in \mathfrak{S}_{\{1,2,3,4\}}$ using one-line notation:

$$\sigma = \big(\sigma(1), \sigma(2), \sigma(3), \sigma(4)\big) = (3, 2, 4, 1) \tag{1}$$

We can similarly write the inverse permutation $\sigma^{-1}$ which maps $\sigma$ back to the previous ordering:

$$\sigma^{-1} = \big(\sigma^{-1}(1), \sigma^{-1}(2), \sigma^{-1}(3), \sigma^{-1}(4)\big) = (4, 2, 1, 3) \tag{2}$$

An *inversion* is a pair $(\sigma(i), \sigma(j))$ such that $i < j$ and $\sigma(i) > \sigma(j)$. We denote $\sigma_S^{\mathrm{id}}$ as the unique permutation within $\mathfrak{S}_S \ \forall S \subseteq N$ with 0 inversions, i.e. the identity permutation. Let $\sigma_{+i,\ell} = \big(\sigma(1), \ldots, \sigma(\ell-1), i, \sigma(\ell+1), \ldots, \sigma(|S|)\big)$ denote the permutation $\sigma \in \mathfrak{S}_{S \setminus \{i\}}$ with $i$ inserted at position $\ell$. The goal of *feature attribution* is to assign a scalar value to each feature $i \in N$ representing its relevance to the model output. In this work, we focus on the Shapley value framework; we introduce Shapley values in §3.1 and discuss how they are used for feature attribution in §3.2.

### 3.1 Shapley Values and Extensions

The Shapley value [65] is a game-theoretic method to attribute the contributions of individual "players" to a cooperative game. Let $\nu : \mathcal{P}(N) \to \mathbb{R}$ be a set function, i.e. the *characteristic function*, that represents the "payoff" assigned to a subset of players $S \subseteq N$, where $\mathcal{P}(N)$ denotes the power set of $N$. The Shapley value is then the unique attribution for $(N, \nu)$ that satisfies several desirable axioms: *Efficiency*, *Symmetry*, *Null Player*, and *Additivity* (App. A.4).

$$\phi_i(N, \nu) = \sum_{S \subseteq N \setminus \{i\}} \frac{(|N| - |S| - 1)!(|S|!)}{|N|!} \left[\nu\left(S \cup \{i\}\right) - \nu(S)\right] \tag{3}$$

The notation $|\cdot|$ indicates set cardinality. Intuitively, the Shapley value averages the *marginal contribution* of feature $i$ (i.e. $\nu\left(S \cup \{i\}\right) - \nu(S)$) over the all subsets $S \subseteq N \setminus \{i\}$. Note that Eq. 3 does not assume any player ordering. A number of extensions have been proposed in this direction [53, 63]; in particular, Sanchez and Bergantiños [63] propose a value that also captures permutations for each subset $S \subseteq N$. Let $\omega : \Omega \to \mathbb{R}$ denote a generalized characteristic function, where $\Omega$ is the set of permutations $\sigma \in \mathfrak{S}_S \ \forall S \subseteq N$. We can then define the Sanchez-Bergañtinos (SB) value:

$$\phi_i^{\mathrm{SB}}(N, \omega) = \sum_{S \subseteq N \setminus \{i\}} \sum_{\sigma \in \mathfrak{S}_S} \frac{(|N| - |S| - 1)!}{|N|!(|S| + 1)} \sum_{l=1}^{|S|+1} \left[\omega\big(\sigma_{+i,\ell}\big) - \omega(\sigma)\right] \tag{4}$$

The SB value has been investigated in the context of network theory [18, 49] and other game theory applications [48], though it has not been investigated in the XAI literature. Similar to the Shapley value, the SB value is the unique value to satisfy the axioms of *Efficiency\**, *Symmetry\**, *Null Player\**, and *Additivity\**, which have been adapted for ordered coalitions (App. A.5).

## 3.2 Shapley Values for Feature Attribution

Shapley values have become widely adopted in machine learning as principled approach for calculating feature attributions in black-box models [61]. Specifically, we define features as "players" in a cooperative game, then define a characteristic function $\nu_{f,x}$ that maps a given sample $x \in \mathcal{X}$, with features $N \setminus S$ ablated, to the output of model $f$. Several such functions have been investigated in the literature [43, 75, 14], differing mainly in their ablation mechanisms. Feature ablation is generally performed by masking ablated features with some value. For example, we can replace features $N \setminus S$ by sampling a distribution $\mathcal{X}$, where $\mathcal{X}$ can be the data distribution or a reference distribution.

$$\nu_{f,x}(S) = \mathbb{E}_{x' \sim \mathcal{X}} \left[ f(x') \mid x'_i = x_i \, \forall i \in S \right] \tag{5}$$

We omit the subscripts $f, x$ when clear from context. In contrast, the SB value have not been investigated in the context of feature attribution. In the next section we discuss challenges with the SB value and introduce a novel framework for attributing importance to feature positions.

# 4 Disentangling Feature Position Importance

Assume that we have a sample $x \in X$ and a prediction model $f$ whose output $f(x)$ depends on the ordering of features in $x$. Our goal is to generate an attribution for each feature in $x$ based on 1) the feature's value, and 2) the feature's position within the sequence. For convenience, we assume that each feature can be permuted to any position within the sequence; in App. B.1 we extend this assumption for applications with fixed subsequences of features or irregular time intervals. We first introduce a novel characteristic function to allow feature permutations (§4.1). We then define OrdShap in §4.2. Approximation methods are addressed in §5.

## 4.1 Quantifying the Effects of Feature Ordering

We propose to quantify the model's output when permuting each feature $i$ across different positions in the sequence. Intuitively, the Shapley value evaluates the model $f$ for every possible subset of features $S \subseteq N$. For each subset $S \subseteq N$, we also want to evaluate $f$ when each feature $i \in S$ is permuted to every position $\ell \in N$. This poses a challenge when applying permutations $\sigma \in \mathfrak{S}_S$ to the sequence $x$, since the features in $N \setminus S$ are not permuted. More formally, for a given feature $i$, there exist positions $\ell, \ell' \in N$ such that:

$$\sum_{\substack{S \subseteq N \\ i \in S}} \left| \{\sigma \in \mathfrak{S}_S : \sigma^{-1}(i) = \ell\} \right| \neq \sum_{\substack{S \subseteq N \\ i \in S}} \left| \{\sigma \in \mathfrak{S}_S : \sigma^{-1}(i) = \ell'\} \right| \tag{6}$$

The proof is detailed in App. C.1. Therefore, feature $i$ is not permuted uniformly across positions. For example, in the singleton set $S = \{i\}$, the only permutation $\sigma \in \mathfrak{S}_S$ is $\sigma_S^{\mathrm{id}}$; therefore $i$ is not permuted to positions $\ell \in N \setminus S$. We therefore define the characteristic function $\tilde{\omega} : \mathcal{P}(N) \times \mathfrak{S}_N \to \mathbb{R}$, which instead rearranges $x$ for a permutation $\sigma \in \mathfrak{S}_N$, then ablates the features in $N \setminus S$:

$$\tilde{\omega}_{f,x}(S, \sigma) = \mathbb{E}_{x' \sim \mathcal{X}} \left[ f(x') \,\middle|\, x'_{\sigma^{-1}(i)} = \begin{cases} x_i & \forall i \in S \\ x'_i & \forall i \in N \setminus S \end{cases} \right] \tag{7}$$

Note that Eq. 7 is a generalization of Eq. 5, reducing to the latter under the identity permutation. Concretely, $\tilde{\omega}_{f,x}(\sigma_N) = \nu_{f,x}(N)$ recovers the model output for sample $x$ under the original order.

## 4.2 OrdShap: Shapley Value with Positional Conditioning

We now define a new attribution $\gamma : \mathbb{N} \times \Theta \to \mathbb{R}^d \times \mathbb{R}^d$, where $\Theta$ is the space of functions $\tilde{\omega}$.

**Definition 1.** *(OrdShap) Given a set of players $N = \{1, \ldots, d\}$ and function $\tilde{\omega} : \mathcal{P}(N) \times \mathfrak{S}_N \to \mathbb{R}$, then the OrdShap values $\gamma_{i,\ell}(N, \tilde{\omega})$ for each player $i \in N$ and position $\ell \in N$ is defined as follows:*

$$\gamma_{i,\ell}(N, \tilde{\omega}) = \sum_{\substack{S \subseteq N \\ i \in S}} \sum_{\substack{\sigma \in \mathfrak{S}_N \\ \sigma^{-1}(i) = \ell}} \frac{(|S| - 1)!(|N| - |S|)!}{(|N| - 1)!|N|!} \left[ \tilde{\omega}(S, \sigma) - \tilde{\omega}(S \setminus \{i\}, \sigma) \right] \tag{8}$$

Definition 1 yields a $d \times d$ matrix, where each row corresponds to a feature $i \in N$, and each column to a position $\ell \in N$. Each element $\gamma_{i,\ell}$ represents the importance of feature $i$, conditioned on being permuted to position $\ell$ in the sequence. The diagonal elements $\gamma_{i,i}$ correspond to the importance of each feature $i$ at its original position, with the remaining features randomly permuted. Thus, OrdShap provide a structured representation of how feature position influences the model output.

Next, while OrdShap offer a detailed view of the position-specific importance, we propose two summary measures: importance related to feature *value* (OrdShap-VI) and *position* (OrdShap-PI).

**OrdShap-VI.** We can disentangle the effect of a feature's value from its position by averaging over all possible positions where that feature might appear (i.e., averaging over $\sigma^{-1}(i)$). This marginalization yields a single attribution score for each feature $i$ that reflects only the feature's contribution due to its value, separate of any positional dependency within the sequence.

$$\bar{\gamma}_i(N, \tilde{\omega}) = \frac{1}{|N|} \sum_{\ell \in N} \gamma_{i,\ell}(N, \tilde{\omega}) \tag{9}$$

This can be interpreted as the Shapley value averaged over all permutations of the features in $x$. Under certain conditions, Eq. 9 has an additional interpretation as the SB value for feature $i$:

**Theorem 1.** *Given a set of players $N = \{1, \dots, d\}$ and a characteristic function $\tilde{\omega} : \mathcal{P}(N) \times \mathfrak{S}_N \to \mathbb{R}$, there exists a corresponding function $\omega : \Omega \to \mathbb{R}$, representing $\tilde{\omega}$ averaged over the permutations $\sigma \in \mathfrak{S}_N$ which contain a given $\pi \in \mathfrak{S}_S \forall S \subseteq N$. Then $\bar{\gamma}_i(N, \tilde{\omega}) = \phi^{SB}(N, \omega)$ is the unique value to satisfy the SB axioms of Efficiency, Symmetry, Null Player, and Additivity (App. A.5).*

The proof is detailed in App. C.2. Further, if we assume that permuting ablated features, which represent noninformative features, to different positions within $x$ has no effect on the model output, then $\bar{\gamma}_i$ corresponds directly to the SB value for feature $i$.

**OrdShap-PI.** We propose a summary measure that quantifies the average change in $\gamma_{i,\ell}$ as the feature position $\ell$ varies, by approximating the positional effects through a linear model. Intuitively, OrdShap-PI provides an interpretable answer to the question: is a given feature more important if it appears later in the sequence, and by how much more? The OrdShap-PI $\beta_i$ is estimated by minimizing the squared error between the centered feature importance $\gamma_{i,\ell} - \bar{\gamma}_i$ and the centered positions $\ell - \bar{\ell}$, where $\bar{\ell} = \sum_{j \in N} j/|N|$:

$$\underset{\beta_i}{\arg \min} \left[ (\ell - \bar{\ell})\beta_i - (\gamma_{i,\ell}(N, v) - \bar{\gamma}_i) \right]^2 \tag{10}$$

This results in an attribution whose magnitude indicates the strength of the positional effect, and whose sign indicates whether the model output increases ($\beta_i > 0$) or decreases ($\beta_i < 0$) as feature $i$ is permuted to later positions in the sequence. Together, the OrdShap-VI values capture the feature importance when averaging over all positions, and OrdShap-PI reflects the average change in model output as a feature's position deviates from its mean position. In practice, a user can leverage the full OrdShap matrix for detailed positional effects on model predictions, or use the OrdShap-VI and OrdShap-PI values for summarized attributions for feature value and position.

### 4.3 Comparison: Shapley, OrdShap, OrdShap-VI, and OrdShap-PI

In Fig. 3 we present a toy example illustrating the differences between the introduced measures. Items are drawn from a pile with replacement, each with a value defined in Table 1. In particular, note that bags have no inherent value, however, any items drawn *before* drawing a bag are discarded, and therefore have value 0. Only *pairs* of gloves, and not individual gloves, have value. We investigate a sample sequence of six items, $x =$[Hat, Hat, Hat, Bag, R-Glove, R-Glove], with a combined value of $\tilde{\omega}_x(\sigma_N^{\text{id}}) = 0$.

Table 1: Payoffs for the toy example.

| Item | Value |
|---|---|
| Bag, L-Glove, R-Glove | 0 |
| Hat | 3 if drawn after "Bag", otherwise 0. |
| Pair of Gloves | 2 if drawn after "Bag", otherwise 0. |

We observe that Shapley Values (§3.2) yields zero attribution for each feature, as the method does not capture order-dependent effects. In contrast, the OrdShap values (Fig. 3B) indicate that several features have non-zero importance but they are highly affected by sequence position. In particular, the "Bag" feature (red) contributes less when it appears later, as fewer items can follow it. The three "Hat" features (blue, orange, green) have equal, increased value in later positions, while the two "R-Glove"

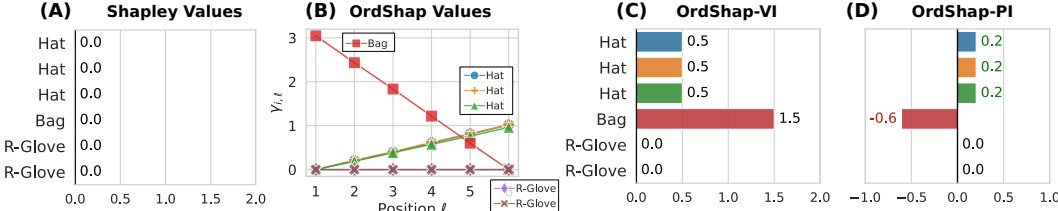

Figure 3: Toy example illustrating the differences between (A) Shapley Values, (B) OrdShap, (C) OrdShap-VI, and (D) OrdShap-PI. Values are calculated on the sample [Hat, Hat, Hat, Bag, R-Glove, R-Glove], with characteristic function defined in §4.3.

features yield no value, as they never form a pair with a "L-Glove" feature. The OrdShap-VI values (Fig. 3C) provide the average contribution across all permutations, isolating feature importance from positional effects. The OrdShap-PI (Fig. 3D) then quantifies position importance; the strongly negative coefficient for "Bag" indicating reduced value in later positions. In summary, OrdShap shows that "Hat" and "Bag" do contribute to the sequence, but their impact is masked by their positional dependency.

## 5 OrdShap Approximation

While OrdShap provides a way to quantify importance due to feature value and position, the formulation in Eq. 8 is generally infeasible due to the $2^d$ subsets and $d!$ permutations. We therefore introduce two model-agnostic, fixed-sample algorithms for approximating OrdShap. §5.1 introduces a sampling-based approach that approximates $\gamma_{i,\ell}$. §5.2 introduces a least-squares approach for directly estimating OrdShap-VI and OrdShap-PI.

### 5.1 Sampling Approximation

We use a permutation sampling approach [74] to approximate $\gamma_{i,\ell}(N, \tilde{\omega})$ with a fixed sample size (detailed in App. B.3, Alg. 1). Our approach samples subsets $S \subseteq N$ and permutations $\sigma \in \mathfrak{S}_N$, then evaluates the marginal contribution to model output when feature $i$ is included in $S$. For a given $\ell \in N$ we filter permutations such that $\sigma^{-1}(i) = \ell$.

### 5.2 Least-Squares Approximation

While the sampling algorithm in §5.1 estimates each $\gamma_{i,\ell}(N, \tilde{\omega})$ using a fixed number of samples, we can further improve efficiency by directly approximating OrdShap-VI and OrdShap-PI using a least-squares (LS) approach. As compared with the sampling algorithm, the LS approach reduces the number of calls to the model $f$, since attributions are calculated for all features simultaneously.

**Definition 2.** *(Least-Squares Approximation of OrdShap) Given a set of players $N = \{1, \ldots, d\}$ and a characteristic function $\tilde{\omega} : \mathcal{P}(N) \times \mathfrak{S}_N \to \mathbb{R}$, then OrdShap-VI and OrdShap-PI are the solutions $\alpha_i$, $\beta_i$, respectively, $\forall i \in N$ for the following minimization problem:*

$$\min_{\alpha_1,\ldots,\alpha_n,\beta_1,\ldots,\beta_n} \sum_{\substack{S \subseteq N \\ S \neq \varnothing, N}} \sum_{\sigma \in \mathfrak{S}_N} \mu(|S|) \left[ \sum_{i \in S} \alpha_i + \sum_{i \in S} \left[ \sigma^{-1}(i) - \bar{\ell} \right] \beta_i - \left[ \tilde{\omega}(S, \sigma) - \tilde{\omega}(\varnothing, \sigma_N^{id}) \right] \right]^2$$

$$s.t. \quad \sum_{i \in N} \alpha_i = \frac{1}{|N|!} \sum_{\sigma \in \mathfrak{S}_N} \tilde{\omega}(N, \sigma) + \tilde{\omega}(\varnothing, \sigma_N^{id}) \quad (11)$$

The coefficient $\mu$ applies a weighting based on subset size. Per Thm. 2 (proof details in App. C.3), setting this coefficient to $\frac{|N|-1}{\binom{|N|}{s}|S|(|N|-s)}$ recovers the SB value for the optimal $\alpha$ coefficients.

**Theorem 2.** *Given a set of players $N = \{1, \ldots, d\}$, characteristic function $\tilde{\omega} : \mathcal{P}(N) \times \mathfrak{S}_N \to \mathbb{R}$, and weighting $\mu(s) = \frac{|N|-1}{\binom{|N|}{s}|S|(|N|-s)}$, then the coefficients $\alpha_1, \ldots, \alpha_N$ minimizing Eq. 11 are SB values for the game $(N, \omega)$, where $\omega$ is the corresponding function defined in Thm. 1.*

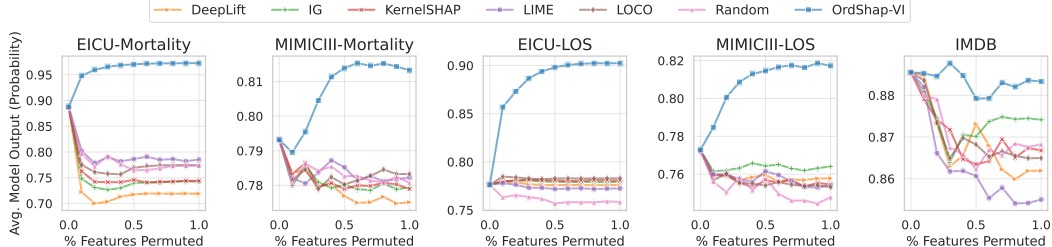

Figure 4: Evaluation of OrdShap-PI (blue) by permuting an increasing number of features and calculating the change in the predicted probability of the predicted class (higher is better). On average, permutating features according to the OrdShap-PI attributions increases or maintains the model's prediction, in contrast to existing methods. AUC calculations and error bars are provided in App. E.1.

The optimal $\beta$ coefficients then represent the linear approximation of the average change in $\gamma_{i,\ell}$ as the position $\ell$ changes, i.e. the OrdShap-PI values. We now approximate Eq. 11 using a fixed number of samples. Let $\mathcal{U}(\mathfrak{S}_N)$ and $\mathcal{U}(\mathcal{P}(N))$ be the uniform distributions over $\mathfrak{S}_N$ and $\mathcal{P}(N)$, respectively. For a given $K, L \in \mathbb{N}$, we draw $K$ samples $S_1, \ldots, S_K \sim \mathcal{U}(\mathcal{P}(N))$ and $KL$ samples $\sigma_1, \ldots, \sigma_{KL} \sim \mathcal{U}(\mathfrak{S}_N)$. We take advantage of two key properties to improve efficiency. First, from the proof of Thm. 2, we can calculate $\alpha$ independently of $\beta$. Second, extending a result from [63], we establish the following corollary (proof details in App. C.4):

**Corollary 2.1.** *Let $\bar{\nu}$ be a characteristic function s.t. $\bar{\nu}(S) = \frac{1}{|S|!} \sum_{\sigma \in \mathfrak{S}_S} \tilde{\omega}(S, \sigma) \, \forall S \subseteq N$. Then the OrdShap-VI value $\bar{\gamma}_i$ is equal to the Shapley value under $\bar{\nu}$, i.e. $\bar{\gamma}_i(N, \tilde{\omega}) = \phi_i(N, \bar{\nu})$.*

This result allows us to solve for $\alpha$ separately using the KernelSHAP algorithm [43]; the solution to $\alpha$ is then used to calculate $\beta$. Let $\Lambda^{(\alpha)} \in \{0, 1\}^{K \times d}$ and $\Lambda^{(\beta)} \in \mathbb{N}^{KL \times d}$ be defined as follows:

$$\Lambda^{(\alpha)}_{i,j} = \mathbb{1}_{S_i}(j) \qquad \Lambda^{(\beta)}_{i,j} = \mathbb{1}_{S_{i \bmod L}}(j)(\sigma_i^{-1}(j) - \bar{\ell}) \tag{12}$$

where "mod" denotes the modulo operator and $\mathbb{1}$ denotes the indicator function. Additionally, let $W^{(\alpha)} \in [0, 1]^{K \times K}$ and $W^{(\beta)} \in [0, 1]^{KL \times KL}$ be diagonal weighting matrices, with diagonal elements $W^{(\alpha)}_{i,i} = \mu(S_i)$, $W^{(\beta)}_{i,i} = \mu(S_{i \bmod L})$ and all other elements equal to 0. Let $F^{(\alpha)} \in \mathbb{R}^K$, $F^{(\beta)} \in \mathbb{R}^{KL}$ represent the model outputs, with elements $F^{(\alpha)}_i = \frac{1}{L} \sum_{l=0}^{L-1} \tilde{\omega}_{x,f}(S_i, \sigma_{lK+i}) - \tilde{\omega}_{x,f}(\varnothing, \sigma_N^{\mathrm{id}})$ and $F^{(\beta)}_i = \tilde{\omega}_{x,f}(S_{i \bmod L}, \sigma_i) - \tilde{\omega}_{x,f}(\varnothing, \sigma_N^{\mathrm{id}})$. It then follows that $\alpha, \beta$ has the solution:

$$\alpha = \left[ (\Lambda^{(\alpha)})^{\intercal} W^{(\alpha)} \Lambda^{(\alpha)} \right]^{-1} (\Lambda^{(\alpha)})^{\intercal} W^{(\alpha)} F^{(\alpha)} \tag{13}$$

$$\beta = \left[ (\Lambda^{(\beta)})^{\intercal} W^{(\beta)} \Lambda^{(\beta)} \right]^{-1} (\Lambda^{(\beta)})^{\intercal} W^{(\beta)} \left[ F^{(\beta)} - \begin{bmatrix} \alpha \\ \vdots \\ \alpha \end{bmatrix} \right] \tag{14}$$

Similar to Lundberg and Lee [43], we enforce the constraint in Eq. 11 by eliminating a feature $j$ in Eq. 13 then solving $\alpha_j$ using the constraint. The algorithm is summarized in App. B.3 Alg. 2. In practice, we can improve efficiency by reducing the number of features to evaluate: for example, including a preliminary feature selection step, combining features into interpretable groups (similar to superpixels [60]), or omitting irrelevant features.

In summary, naively evaluating Eq. 8 incurs $\mathcal{O}(d! 2^d \delta_f)$ complexity, where $\delta_f$ is the cost of evaluating the model $f$. We reduce this to $\mathcal{O}(dKL\delta_f + d^2KL)$ with the sampling algorithm, and $\mathcal{O}(KL\delta_f + d^2KL + d^3)$ with the LS algorithm. In practice, the LS algorithm is typically faster than the Sampling algorithm since function evaluations are often the performance bottleneck. However, the LS algorithm only calculates OrdShap-PI and OrdShap-VI values, therefore the Sampling algorithm is required to calculate the full OrdShap value matrix in Def. 1.

## 6 Experiments

We empirically evaluate OrdShap on its ability to identify the importance of a given sample's feature ordering. In §6.1 we quantitatively evaluate OrdShap against existing attribution methods. In §6.2

Table 2: Evaluation of OrdShap-VI using Inclusion AUC (higher is better) and Exclusion AUC (lower is better) metrics, with standard error in parentheses. The best-performing explainer for each model is bolded. The associated curves used to generate these results are provided in App. E.1.

| Metric | Explainer | EICU-LOS | EICU-Mort | MIMICIII-LOS | MIMICIII-Mort | IMDB |
|---|---|---|---|---|---|---|
| Inclusion AUC ↑ | DL | 0.906 (0.010) | 0.804 (0.013) | 0.893 (0.011) | 0.854 (0.012) | 0.784 (0.016) |
| | IG | 0.903 (0.010) | 0.803 (0.013) | 0.896 (0.011) | 0.859 (0.012) | 0.791 (0.016) |
| | KS | 0.904 (0.010) | 0.801 (0.014) | 0.898 (0.011) | 0.855 (0.012) | 0.863 (0.011) |
| | LIME | 0.866 (0.012) | 0.776 (0.016) | 0.885 (0.012) | 0.804 (0.015) | 0.859 (0.012) |
| | LOCO | 0.904 (0.010) | 0.799 (0.014) | 0.894 (0.011) | 0.854 (0.012) | 0.855 (0.012) |
| | Random | 0.812 (0.014) | 0.769 (0.016) | 0.851 (0.014) | 0.705 (0.017) | 0.797 (0.016) |
| | OrdShap-VI | **0.913** (0.010) | **0.809** (0.013) | **0.899** (0.011) | **0.862** (0.011) | **0.866** (0.012) |
| Exclusion AUC ↓ | DL | 0.614 (0.025) | 0.727 (0.022) | 0.817 (0.018) | 0.484 (0.025) | 0.855 (0.012) |
| | IG | 0.629 (0.024) | 0.728 (0.021) | 0.795 (0.019) | 0.481 (0.025) | 0.867 (0.013) |
| | KS | 0.626 (0.024) | 0.730 (0.021) | 0.805 (0.019) | 0.485 (0.025) | **0.766** (0.018) |
| | LIME | 0.771 (0.017) | 0.750 (0.018) | 0.840 (0.016) | 0.591 (0.022) | 0.804 (0.016) |
| | LOCO | 0.618 (0.025) | 0.732 (0.021) | 0.803 (0.019) | 0.495 (0.025) | 0.784 (0.017) |
| | Random | 0.821 (0.014) | 0.759 (0.017) | 0.866 (0.013) | 0.694 (0.018) | 0.849 (0.014) |
| | OrdShap-VI | **0.573** (0.028) | **0.724** (0.022) | **0.788** (0.020) | **0.472** (0.025) | 0.779 (0.021) |

we investigate OrdShap on a synthetic dataset. In §6.3 we qualitatively compare OrdShap with KernelSHAP. In §6.4 we present execution time results. All experiments were performed on an internal cluster using AMD 7302 16-Core processors and NVIDIA A100 GPUs.

**Datasets and Models.** We evaluated OrdShap on two EHR datasets (MIMICIII [34] and EICU [57]) and a natural language dataset (IMDB [45]). Full dataset and model details are provided in App. D.1. MIMICIII and EICU are large-scale studies for Intensive Care Unit (ICU) patient stays. Both datasets include sequences of clinical events for each patient stay, such as administered lab tests, infusions, and medications. We follow Hur et al. [31] for data preprocessing and tokenization, then train BERT classification models [19] on two tasks for each dataset: 1) Mortality and 2) Length-of-Stay $\geq 3$ days (LOS). We also include the IMDB dataset, containing over 50,000 movie reviews, on a sentiment analysis task. We apply a pretrained DistilBERT model [64] using the Huggingface library [85], then evaluate the relevance of each sentence towards the predicted sentiment of the entire movie review.

**Explainers.** Unless otherwise stated, we use the OrdShap Least-Squares algorithm (§5.2) due to computational constraints. We compare OrdShap to a variety of attribution methods: KernelSHAP (KS) [43], LIME [60], and LOCO [40] are perturbation-based approaches. We additionally compare against gradient-based approaches: Integrated Gradients (IG) [76], and DeepLIFT (DL) [66]. We also include a baseline, Random, consisting of attributions drawn from $\mathcal{U}(0,1)^d$. Additional detail on the implemented methods are provided in App. D.2.

## 6.1 Quantitative Evaluation

We compare OrdShap to a variety of attribution methods to evaluate how (1) OrdShap-PI captures feature position importance, and (2) OrdShap-VI calculates an attribution that incorporates position information. For all explainers, we generate attributions for 200 test samples which are evaluated using quantitative metrics.

**(1) Position Importance.** The OrdShap-PI attributions quantify both the magnitude and direction of positional impact on feature importance with respect to model predictions. Therefore, to evaluate OrdShap-PI, we systematically permute samples according to the generated attributions and measure the change in model output. Specifically, we select a subset of features based attribution magnitude, then permute features toward the beginning (negative attributions) or end (positive attributions) of the sequence. We then measure the model output probability for the predicted class, $P(Y = \hat{y}|\tilde{x})$, where $\tilde{x}$ represents the permuted sample and $\hat{y}$ is the predicted class under the original sample $x$. We compare to competing methods by similarly permuting features according to their attributions.

Results are presented in Fig. 4. As expected, we observe that permuting features according to OrdShap-PI generally increases the model output across feature subsets. Interestingly, the model output remains relatively flat for IMDB when permuting features; this suggests that the model is able to accurately predict sentiment regardless of sentence order. In contrast, competing methods fail to capture the importance of feature ordering. Permuting features according to traditional attributions either does not affect (EICU-LOS) or negatively affects (all other models) affects model output.

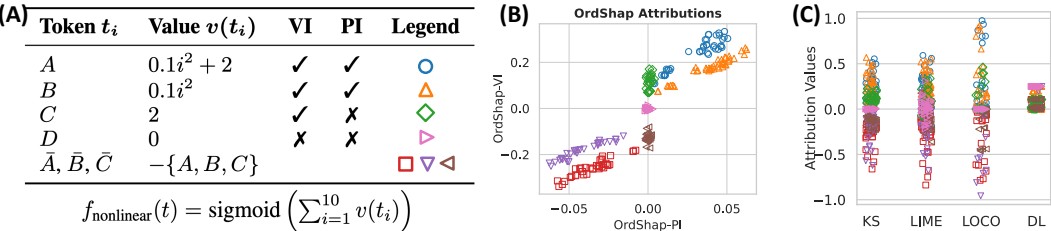

Figure 5: Attributions for a synthetic dataset and model $f_{\text{nonlinear}}$. **(A)** Token values; tokens are assigned Value Importance (VI) and/or Position Importance (PI) with respect to sequence index $i$. **(B)** OrdShap-VI and OrdShap-PI attributions are able to separate the different tokens based on VI and PI effects. **(C)** In contrast, attributions from existing methods cannot distinguish between the different tokens since VI and PI effects are entangled.

**(2) Value Importance.** OrdShap-VI attributions quantify a feature's contribution to model predictions independent of positional effects, making them robust across different sequence orderings. To evaluate this property, we extend the Inclusion/Exclusion AUC (IncAUC / ExcAUC) [33] metrics to incorporate feature permutations (details provided in App. D.3). More concretely, for each sample we rank features by attribution scores and iteratively select the top $k\%$ of features. These features are then either masked from the original sample (ExcAUC) or retained in an otherwise completely masked sample (IncAUC). Importantly, we add a permutation step that reorders features in the masked sample. We then measure agreement (*post-hoc accuracy*) between the model's predictions under the masked, permuted samples and original sample, averaging over 10 permutations for each $k$, then summing the area under the resulting curve for different $k$ values. We additionally implement a similarly modified iAUC/dAUC metric [56], which directly evaluates the model output rather than post-hoc accuracy.

IncAUC/ExcAUC results are shown in Table 2 (iAUC/dAUC results in App. E.1 Table 5). We observe that OrdShap-VI outperforms all competing methods on IncAUC, and performs well on ExcAUC, only second to KernelSHAP on the IMDB dataset. This supports our observation from (1), that the DistilBert model predictions are only mildly affected by sentence order in IMDB. Overall the results indicate that OrdShap-VI provides a more robust attribution under feature permutation, and better captures each feature's *value importance* independent of positional context.

## 6.2 Attribution Disentanglement on Synthetic Data

We create a synthetic dataset to evaluate the disentanglement of value and position importance. We sample 200 sequences $t^{(n)}$, $n \in \{1, \ldots, 200\}$, of length 10, where each element $t_i^{(n)}$ is a token sampled uniformly from the set $\{A, B, C, D, \bar{A}, \bar{B}, \bar{C}\}$. We define two models: $f_{\text{linear}}(t) = \sum_{i=1}^{10} v(t_i)$ and $f_{\text{nonlinear}}(t) = \text{sigmoid}(f_{\text{linear}}(t))$, where $v(t_i)$ is the value of token $t_i$ (Fig. 5A).

Fig. 5 shows the results for $f_{\text{nonlinear}}$. We observe that the associated OrdShap-VI and OrdShap-PI (Fig. 5B) follow the expected VI and PI values from Fig. 5A. In particular, tokens $A, B, \bar{A}, \bar{B}$ exhibit nonzero OrdShap-PI values, which indicates dependency on feature position. In contrast, tokens $C, D, \bar{C}$ exhibit OrdShap-PI $\approx 0$, indicating no positional dependency. In addition, OrdShap-VI and OrdShap-PI attributions are able to fully separate the 7 tokens according to their value and position importance. This contrasts with traditional attribution methods (Fig. 5C), in which value and position importance are entangled together, and thus the tokens are unable to be separated.

The results for $f_{\text{linear}}$ are shown in App. E.3 Fig. 9. The additive nature of $f_{\text{linear}}$ results in constant OrdShap attributions for all samples in the dataset. However, existing metrics (Fig. 9C) are still unable to disentangle the effects of value and position importance.

## 6.3 Case Study on Medical Tokens

In Fig. 6A we further investigate the LOS $\geq 3$ prediction sample from Fig. 1. In this sample, the model predicts LOS $< 3$ days, therefore positive attributions indicate important features with respect to LOS $< 3$ days. Several key observations emerge. Many medication tokens have high OrdShap-VI values, which suggests that these medications correspond to LOS $< 3$ prediction. In particular, Potassium Chloride is generally used to treat potassium deficiency, a relatively minor condition that does not generally require a long hospital stay. OrdShap-PI (negative with low magnitude) indicates

Table 3: Execution time results, in seconds, averaged over 200 samples. Standard deviation results are provided in parentheses. We compare existing attribution methods with OrdShap, calculated using the Least Squares (LS) and Sampling (S) algorithms. Note that competing methods cannot disentangle importance due to feature value and position.

| Explainer | EICU-LOS | EICU-Mort | MIMICIII-LOS | MIMICIII-Mort | IMDB |
|---|---|---|---|---|---|
| DL | 0.02 (0.03) | 0.01 (0.03) | 0.02 (0.03) | 0.01 (0.03) | 0.08 (0.03) |
| IG | 0.13 (0.25) | 0.04 (0.07) | 0.10 (0.08) | 0.03 (0.04) | 0.22 (0.08) |
| KS | 0.69 (0.08) | 0.21 (0.08) | 0.68 (0.10) | 0.17 (0.06) | 1.85 (1.88) |
| LIME | 0.14 (0.04) | 0.11 (0.06) | 0.16 (0.07) | 0.12 (0.05) | 0.14 (0.05) |
| LOCO | 0.57 (0.13) | 0.38 (0.09) | 0.36 (0.16) | 0.26 (0.11) | 0.06 (0.04) |
| OrdShap (LS) | 29.29 (1.30) | 5.53 (0.30) | 27.33 (4.41) | 5.49 (1.23) | 54.61 (72.16) |
| OrdShap (S) | 2470.73 (685.0) | 984.03 (264.0) | 1369.00 (721.7) | 565.51 (288.9) | 680.23 (440.9) |

that this token is less predictive of LOS $< 3$ (and conversely, more predictive of LOS $\geq 3$) when it occurs later in the sequence. We observe that, indeed, KernelSHAP attributions are relatively low compared with OrdShap-VI for this token due to its position, however KernelSHAP fails to distentangle the positional effect.

Interestingly, Bedside Glucose Test is assigned a high KernelSHAP value, but OrdShap indicates that this token's importance is mainly positional; having this test later in the sequence is indicative of longer LOS. This follows intuition, as Bedside Glucose is a common test that does not typically indicate a patient's condition. Again, KernelSHAP assigns this token a high attribution value since it appears within the last quartile of the original sequence, but cannot distinguish between the positional and value importance.

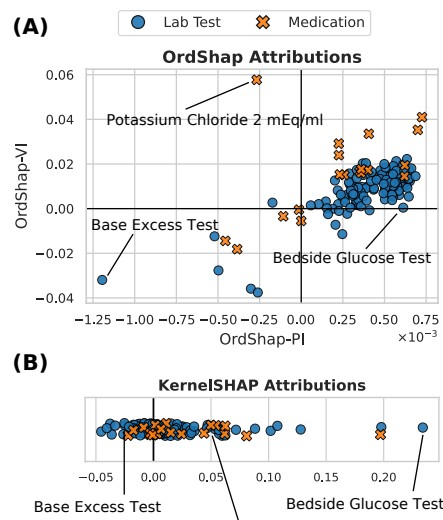

Figure 6: Case study for the sample from Fig. 1. **(A)** OrdShap-PI and OrdShap-VI attributions. **(B)** KernelSHAP attributions.

### 6.4 Execution Time Evaluation

We provide execution time results for OrdShap and competing methods in Table 3. Both OrdShap algorithms are generally slower than competing methods; this is this cost of calculating positional attributions, which requires evaluating the black-box model over multiple permutations of the input sequence. As expected, the the Least Squares algorithm (LS) significantly improves computational efficiency compared with the Sampling algorithm (S), and therefore should be the preferable algorithm when the user does not require calculating the entire $\gamma$ matrix.

## 7 Limitations and Conclusion

While many feature attributions methods exist for sequential deep learning models, existing methods assume a fixed feature ordering when explaining model predictions. Our work addresses this gap by introducing OrdShap, a novel approach that uses permutations to quantify the importance of a feature's position within the input sequence. Regarding limitations, Shapley-based methods are often computationally expensive due to the summation over $2^d$ coalitions; OrdShap additionally requires averaging over $d!$ permutations. We address this limitation through sampling and approximation (§5), and leave further improvements for future work. In particular, many algorithms have been proposed for improving the efficiency of Shapley value approximation (App. A.3) which could possibly be extended to approximate OrdShap. Additionally, while we theoretically establish a relationship between OrdShap-VI and Sanchez-Bergantiños values in this work (Thm. 1), OrdShap-PI requires a different axiomatization based on positional importance, which we leave for future work.

## Acknowledgements

This work was supported in part by the following grants from the National Heart, Lung, and Blood Institute: NIH 2T32HL007427-41, U01HL089856, 5R01HL167072, and 5R01HL171213-02. We thank the NeurIPS reviewers and meta-reviewers for their helpful comments and feedback.

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

# A Background

## A.1 Societal Impact

As machine learning models are increasingly used for decision-making in high-impact domains [58, 15, 28, 87, 10, 9], it is important to develop explainable AI (XAI) methods to improve understanding of model predictions. XAI methods help to increase model transparency and reliability, which allows for informed decision-making when using the prediction models. The proposed method, OrdShap, addresses a critical gap in explainable AI for sequential models by quantifying how feature ordering influences model predictions.

However, we acknowledge that improved explainability tools could potentially be misused to reverse-engineer proprietary models or reconstruct private information from models trained on medical data. Additionally, like all attribution methods, OrdShap provides simplified approximations of complex model behavior that may create overconfidence in model predictions. Practitioners should use these XAI methods to in conjunction with human judgment and combine them with other safeguards. For example, recent works have developed uncertainty quantification methods [68, 29, 16], as well as metrics to evaluate explainer stability [81], robustness [2, 35, 67], and complexity [30, 52, 7]. These additional tools can be used to mitigate the possible negative societal effects of XAI.

## A.2 Machine Learning with Electronic Health Record (EHR) Data

Many of the examples and experiments in the main text use electronic health record (EHR) data, which consist of clinical events that occur during a patient's interactions with hospitals or clinics. Clinical events can include medical diagnoses (e.g. ICD-9 codes), prescribed medications, medical procedures, or laboratory tests. Each sample in the EICU and MIMICIII datasets (see App. D) contains the history for an individual patient, which consists of lists of these clinical events over time. Recent works [59, 41, 55, 27, 31] have explored tokenizing these clinical events for use with Transformers in various prediction tasks. While there are a variety of approaches, most convert each unique clinical event to a separate token, which we refer to as medical tokens in the manuscript. Therefore, the processed EHR datasets in the main text consist of tokenized samples, where each sample represents a separate patient, and each token represents a clinical event.

Below we provide examples of medical tokens. The MIMICIII and EICU datasets include medications, laboratory tests, and infusions.

**Medications:**
- Sodium Chloride 0.9% IV
- Insulin Lispro (Human) 100 unit/mL
- Dextrose 50% IV Solution
- Hydrocortisone Sodium Succinate PF 100 mg
- Magnesium Sulfate 50% Injection Solution

**Laboratory Tests:**
- total bilirubin
- platelets x 1000
- WBC x 1000 [White blood cell count]
- MPV [Mean Platelet Volume]
- Glucose

**Infusions:**
- Normal Saline 20K
- Epinephrine
- Sodium Bicarbonate mL/hr
- Propofol
- Isoproterenol

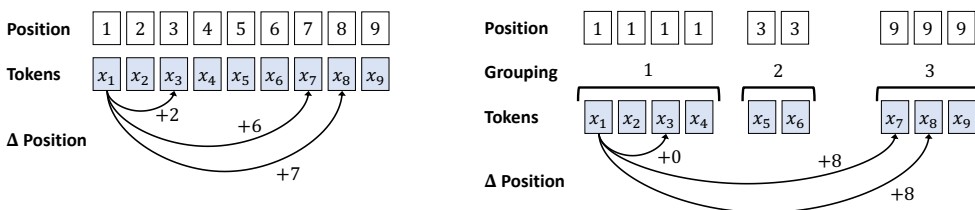

Figure 7: Extension of OrdShap to alternative position indexing. **(A)** A sequence of features, presented as tokens. We assume that each token can be permuted to each position. For convenience, this is the assumed formulation in the main text. **(B)** Alternative sequence of tokens, with tokens separated into permutation-invariant groups and occurring at an independent, irregular time indexing.

## A.3  Background: Shapley Value Approximation

Shapley-based methods are often prohibitively expensive to compute exactly in the general case; many methods been proposed to address this challenge [14]. Some algorithms take advantage of model-specific properties to compute Shapley values, such as for linear models [43, 74], trees [44], or neural networks [43, 21, 3]. Model-agnostic methods generally sample a number of subsets $< 2^d$ at random. These stochastic approaches include permutation sampling methods [73, 74, 11] and also least-squares methods [12, 43]. Several works have further investigated and improved both sampling [50] and least-squares approaches [16, 54, 84]. An alternate line of work has also investigated Shapley approximation approaches for higher-order attributions [38, 24, 25].

## A.4  Shapley Value Axioms [65]

**Efficiency.** $\sum_{i \in N} \phi_i = \nu(N)$.

**Symmetry.** Define $\nu_\sigma$ as the characteristic function $\nu$ where the players are permuted according to permutation $\sigma \in \mathfrak{S}_N$. For any player $i$ and permutation $\sigma \in \mathfrak{S}_N$, $\phi_i(N, \nu_\sigma) = \phi_i(N, \nu)$.

**Null Player.** A player $i$ is considered a *null player* in the game $(N, \nu)$ if for every $S \subseteq N \setminus \{i\}$, $\nu(S) = \nu(S \cup \{i\})$. The *null player axiom* states that for every null player $i$, $\phi_i(N, \nu) = 0$.

**Additivity.** Let $\nu, \dot{\nu}$ be two different characteristic functions. Let $(\nu + \dot{\nu})(S)$ represent the characteristic function $\nu(S) + \dot{\nu}(S) \ \forall S \subseteq N$. Then for any player $i$, $\phi_i(N, \nu + \dot{\nu}) = \phi_i(N, \nu) + \phi_i(N, \dot{\nu})$.

## A.5  Sanchez-Bergantiños Value Axioms [63]

**Efficiency\*.** $\sum_{i \in N} \phi_i^{\text{SB}}(N, \omega) = \frac{1}{|N|!} \sum_{\sigma \in \mathfrak{S}_N} \omega(\sigma)$.

**Symmetry\*.** Two players $i, j$ are *symmetric* in the game $(N, \omega)$ if $\omega(\sigma_{+i,\ell}) = \omega(\sigma_{+j,\ell})$ for all $S \subseteq N \setminus \{i, j\}$, $\sigma \in \mathfrak{S}_S$, and $l \in \{1, \ldots, |S| + 1\}$. The *symmetry axiom* states that for any symmetric players $i, j$, $\phi_i^{\text{SB}}(N, \omega) = \phi_j^{\text{SB}}(N, \omega)$.

**Null Player\*.** A player $i$ is considered a *null player* in the game $(N, \omega)$ if $\omega(\sigma) = \omega(\sigma_{+i,\ell})$ for all $S \subseteq N \setminus \{i\}$, $\sigma \in \mathfrak{S}_S$, and $l \in \{1, \ldots, |S| + 1\}$. The *null player axiom* states that for every null player $i$, $\phi_i^{\text{SB}}(N, \omega) = 0$.

**Additivity\*.** Let $\omega, \dot{\omega}$ be two different characteristic functions. Let $(\omega + \dot{\omega})(\sigma)$ represent the characteristic function $\omega(\sigma) + \dot{\omega}(\sigma) \ \forall \sigma \in \mathfrak{S}_S, S \subseteq N$. Then for any player $i$, $\phi_i^{\text{SB}}(N, \omega + \dot{\omega}) = \phi_i^{\text{SB}}(N, \omega) + \phi_i^{\text{SB}}(N, \dot{\omega})$.

---

**Algorithm 1** OrdShap Sampling Algorithm

---

**Input :** Data sample $x \in \mathbb{R}^d$; Model $f$; Number of Samples K, L.
**Output :** $\gamma \in \mathbb{R}^{d \times d}$

---

$\gamma \leftarrow \mathbf{0}$
**for** $i \in \{1, \ldots, d\}$ **do**
    **for** $l \in \{1, \ldots, L\}$ **do**                                               `// Sample Permutations`
        Sample a permutation $\sigma_l \sim \mathcal{U}(\mathfrak{S}_N)$
        **for** $k \in \{1, \ldots K\}$ **do**                                     `// Sample Subsets`
            Sample a permutation $\sigma_k \sim \mathcal{U}(\mathfrak{S}_N)$
            $S \leftarrow c(\sigma_k, \sigma_k^{-1}(i))$
            $\Delta \leftarrow \tilde{\omega}_{x,f}(S, \sigma_l) - \tilde{\omega}_{x,f}(S \setminus \{i\}, \sigma_l)$
            $\gamma_{i,\sigma_l^{-1}(i)} \leftarrow \gamma_{i,\sigma_l^{-1}(i)} + \Delta$
        **end**
    **end**
**end**

Return $\frac{1}{LK}\gamma$

---

# B   Additional Method Details

## B.1   Generalization to Alternative Position Indexing

In the main text we assume that each feature can be permuted to any position in the sequence (Fig. 7A). However, in some applications, (1) features may be naturally grouped into permutation-invariant sets, or (2) features may occur at irregular time indices (Fig. 7B). For example, in EHR data, a sequence of patient tokens may be grouped into individual visits; in this case, a user may want to enforce the tokens within each visit to be permutation invariant. Visits may also occur at irregular time indices, therefore permuting tokens between visits should not always result in the same change in position.

OrdShap can be extended to accommodate these constraints while maintaining the same framework and theoretical properties. Intuitively, the position index used in the main text can be mapped to any other index, including time. Under this new mapping, for a given feature, OrdShap-PI would represent position importance with respect to units of time rather than position index. OrdShap-VI would correspondingly represent value importance with respect to the mean time of the sequence.

More concretely, let $N = \{1, \ldots, |N|\}$ be the original token indices and let $G$ be the grouped time indices (e.g., visit-level indices). We define a mapping function $g : N \rightarrow G$ that associates each token with its corresponding group. As illustrated in Fig. 7B, we might have $N = \{1, \ldots, 9\}$ and $G = \{1, 3, 9\}$. We can then rewrite Eq. 8 to account for the new constraints:

$$\gamma_{i,\ell}(N, \tilde{\omega}) = \sum_{\substack{S \subseteq N \\ i \in S}} \sum_{\substack{\sigma \in \mathfrak{S}_N \\ (g \circ \sigma^{-1})(i) = \ell}} \frac{(|S| - 1)!(|N| - |S|)!}{(|N| - 1)!|N|!} \left[ \tilde{\omega}(S, \sigma) - \tilde{\omega}(S \setminus \{i\}, \sigma) \right] \quad (15)$$

Note that in Eq. 15, $\ell \in G$ rather than $\ell \in N$ from the original definition of $\gamma$; therefore, this results in a $d \times |G|$ matrix rather than a $d \times d$ matrix. In addition, Eq. 15 still evaluates $\tilde{\omega}$ on each permutation $\sigma \in \mathfrak{S}_N$, however permutations within each group are effectively averaged together, resulting in the desired within-group permutation invariance. Therefore, after this extension, the remainder of the framework and algorithm remain unchanged with position indices $\ell \in G$ representing grouped positions rather than individual position indices.

## B.2   OrdShap Approximation: Sampling Algorithm

In Algorithm 1, we present the sampling algorithm as described in §5.1. We define a function $c$ that takes an ordering $\sigma \in \mathfrak{S}_N$ and a position $i \in \{1, \ldots, N\}$ and returns the set of elements in $\sigma$ that are ordered before $i$; i.e. $\{j : \sigma^{-1}(j) < \sigma(i)^{-1}\}$.

---

**Algorithm 2** OrdShap Least-Squares Algorithm

---

**Input :** Data sample $x \in \mathbb{R}^d$; Model $f$; Number of Samples K, L.
**Output :** $\alpha \in \mathbb{R}^d$, $\beta \in \mathbb{R}^d$

$\Lambda^{(\alpha)}, \Lambda^{(\beta)}, W^{(\alpha)}, W^{(\beta)}, F^{(\alpha)}, F^{(\beta)}, \alpha, \beta \leftarrow \mathbf{0}$                 `// Initialize variables`

Sample permutations $\sigma_1, \ldots, \sigma_{KL} \sim \mathcal{U}(\mathfrak{S}_N)$             `// Sample Permutations`
Sample subsets $S_1, \ldots, S_K \sim \mathcal{U}(\mathcal{P}(N))$                 `// Sample Subsets`

$\delta \leftarrow \tilde{\omega}_{x,f}(\varnothing, \sigma_N^{\mathrm{id}})$                        `// Compute the baseline`
$\bar{\ell} \leftarrow \frac{1}{|N|} \sum_{j \in N} j$

**for** $i \in \{1, \ldots, KL\}$ **do**
    **for** $j \in \{1, \ldots, d\}$ **do**
        $\Lambda_{i,j}^{(\beta)} \leftarrow \mathbb{1}_{S_{i \bmod L}}(j)(\sigma_i^{-1}(j) - \bar{\ell})$
    **end**
    $W_{i,i}^{(\beta)} \leftarrow \mu(S_{i \bmod L})$
    $F_i^{(\beta)} \leftarrow \tilde{\omega}_{x,f}(S_{i \bmod L}, \sigma_i) - \delta$         `// Evaluate f under permutations and subsets`
**end**

**for** $i \in \{1, \ldots, K\}$ **do**
    **for** $j \in \{1, \ldots, d\}$ **do**
        $\Lambda_{i,j}^{(\alpha)} \leftarrow \mathbb{1}_{S_i}(j)$
    **end**
    $W_{i,i}^{(\alpha)} \leftarrow \mu(S_i)$
    $F_i^{(\alpha)} \leftarrow \frac{1}{L} \sum_{l=0}^{L-1} F_{lK+i}^{(\beta)}$
**end**

$F^{(\alpha)} \leftarrow F^{(\alpha)} - F^{(\alpha)} \odot \Lambda_{:,1}^{(\alpha)}$           `// Remove a feature to enforce the constraint.`

$\alpha_{2:} \leftarrow \left[ (\Lambda_{:,2:}^{(\alpha)})^{\intercal} W^{(\alpha)} \Lambda_{:,2:}^{(\alpha)} \right]^{-1} (\Lambda_{:,2:}^{(\alpha)})^{\intercal} W^{(\alpha)} F^{(\alpha)}$           `// Eq. 13`

$\alpha_1 \leftarrow \frac{1}{L} \sum_{i=1}^{L} \tilde{\omega}(N, \sigma_i) + \tilde{\omega}(\varnothing, \sigma_N^{\mathrm{id}}) - \sum_{j=2}^{d} \alpha_j$           `// Enforce constraint.`

$\beta \leftarrow \left[ (\Lambda^{(\beta)})^{\intercal} W^{(\beta)} \Lambda^{(\beta)} \right]^{-1} (\Lambda^{(\beta)})^{\intercal} W^{(\beta)} \left[ F^{(\beta)} - \begin{bmatrix} \alpha \\ \vdots \\ \alpha \end{bmatrix} \right]$           `// Eq. 14`

    Return $\alpha, \beta$

---

## B.3   OrdShap Approximation: Least-Squares Algorithm

In Algorithm 2, we present the sampling algorithm as described in §5.2.

## C  Theoretical Details

### C.1  Proof of Permutation Non-uniformity.

We want to show that, for a given feature $i$, there exist positions $\ell, \ell' \in N$ such that:

$$\sum_{\substack{S \subseteq N \\ i \in S}} \left|\{\sigma \in \mathfrak{S}_S : \sigma^{-1}(i) = \ell\}\right| \neq \sum_{\substack{S \subseteq N \\ i \in S}} \left|\{\sigma \in \mathfrak{S}_S : \sigma^{-1}(i) = \ell'\}\right| \tag{16}$$

*Proof.* We set up a proof by contradiction. Assume that the following equation holds for all $\ell, \ell' \in N$:

$$\sum_{\substack{S \subseteq N \\ i \in S}} \left|\{\sigma \in \mathfrak{S}_S : \sigma^{-1}(i) = \ell\}\right| = \sum_{\substack{S \subseteq N \\ i \in S}} \left|\{\sigma \in \mathfrak{S}_S : \sigma^{-1}(i) = \ell'\}\right| \tag{17}$$

Combining components in the summations:

$$\sum_{\substack{S \subseteq N \\ i \in S}} \left[\left|\{\sigma \in \mathfrak{S}_S : \sigma^{-1}(i) = \ell\}\right| - \left|\{\sigma \in \mathfrak{S}_S : \sigma^{-1}(i) = \ell'\}\right|\right] = 0 \tag{18}$$

We fix $\ell = i$, i.e. the position of feature $i$. We also fix $\ell' = j$ for a given $j \in N \setminus \{i\}$. Therefore, we can rewrite the equation as follows:

$$\sum_{\substack{S \subseteq N \\ i \in S \\ j \in S}} \Big[\underbrace{\left|\{\sigma \in \mathfrak{S}_S : \sigma^{-1}(i) = \ell\}\right| - \left|\{\sigma \in \mathfrak{S}_S : \sigma^{-1}(i) = \ell'\}\right|}_{=0}\Big]$$

$$+ \sum_{\substack{S \subseteq N \\ i \in S \\ j \notin S}} \Big[\underbrace{\left|\{\sigma \in \mathfrak{S}_S : \sigma^{-1}(i) = \ell\}\right|}_{>0} - \underbrace{\left|\{\sigma \in \mathfrak{S}_S : \sigma^{-1}(i) = \ell'\}\right|}_{=0}\Big] = 0 \quad (19)$$

Note that the first summation in Eq. 19 reduces to zero, since the permutations within $\mathfrak{S}_S$ where $S$ contains both $i$ and $j$ will uniformly permute feature $i$ to positions $\ell$ and $\ell'$. In the second summation, the first term is strictly positive, since $i$ is always in the set $S$. However, since $j \notin S$, there exist no permutations $\sigma \in \mathfrak{S}_S$ that permute feature $i$ to position $\ell'$. Therefore, the second term in the second summation will always be zero. This forms a contradiction, which completes the proof.

$\square$

### C.2  Proof of Theorem 1.

*Proof.* We want to show that, given the value $\bar{\gamma}_i(N, \tilde{\omega})$, there exists a corresponding characteristic function $\omega$ such that $\bar{\gamma}_i(N, \tilde{\omega}) = \phi_i^{\text{SB}}(N, \omega)$.

We first introduce the relevant notation. Given permutations $\pi \in \mathfrak{S}_S$ and $\sigma \in \mathfrak{S}_N$ for $S \subseteq N$, we define $\zeta(\pi, \sigma)$ as the number of pairwise disagreements between $\pi$ and $\sigma$:

$$\zeta(\pi, \sigma) = |\{(\pi(i), \pi(j)) : \pi(i) < \pi(j) \wedge \sigma(i) > \sigma(j)\}| \tag{20}$$

Let $\Gamma(\mathfrak{S}_N, \pi) = \{\sigma \in \mathfrak{S}_N : \zeta(\pi, \sigma) = 0\}$ represent the set of all permutations $\sigma \in \mathfrak{S}_N$ such that there are 0 pairwise disagreements between $\pi$ and $\sigma$. Let $T(\pi) = \{i \in \mathbb{N} : i \in \pi\}$ denote the set containing the elements in permutation $\pi$. We define the characteristic function $\omega : \Omega \to \mathbb{R}$, where $\Omega$ is the set of permutations $\sigma \in \mathfrak{S}_S \ \forall S \subseteq N$:

$$\omega(\pi) = \frac{(|T(\pi)|)!}{|N|!} \sum_{\sigma \in \Gamma(\mathfrak{S}_N, \pi)} \tilde{\omega}\big(T(\pi), \sigma\big) \tag{21}$$

Note that for a given $\pi \in \mathfrak{S}_S$, $|\Gamma(\mathfrak{S}_N, \pi)| = \frac{|N|!}{|S|!}$. The characteristic function $\omega$ represents $\tilde{\omega}$ averaged over all permutations in $\mathfrak{S}_N$ with the relative order of elements in $\pi$ fixed, and $N \setminus S$ elements masked.

For a given permutation $\sigma \in \mathfrak{S}_S$, let $\sigma_{-i}$ denote the permutation $\sigma$ with $i$ removed, i.e. $\sigma_{-i} = (\sigma_1, \ldots, \sigma_{k-1}, \sigma_{k+1}, \ldots, \sigma_{|S|})$ such that $k = \sigma^{-1}(i)$. Additionally, let $\sigma_{+i,k}$ represent $\sigma$ with $i$ inserted at index $k$, i.e. $\sigma_{+i,k} = (\sigma_1, \ldots, \sigma_{k-1}, i, \sigma_{k+1}, \ldots, \sigma_{|S|})$.

We now begin the proof. From Eq. 8:

$$\gamma_{i,\ell}(N, \tilde{\omega}) = \sum_{\substack{S \subseteq N \\ i \in S}} \sum_{\substack{\sigma \in \mathfrak{S}_N \\ \sigma^{-1}(i) = \ell}} \frac{(|S|-1)!(|N|-|S|)!}{(|N|-1)!|N|!} \left[ \tilde{\omega}(S, \sigma) - \tilde{\omega}(S \setminus \{i\}, \sigma) \right]$$

Averaging over $\ell \in \{1, \ldots, |N|\}$:

$$\frac{1}{|N|} \sum_{\ell=1}^{|N|} \gamma_{i,\ell} = \frac{1}{|N|} \sum_{\ell=1}^{|N|} \sum_{\substack{S \subseteq N \\ i \in S}} \sum_{\substack{\sigma \in \mathfrak{S}_N \\ \sigma^{-1}(i) = \ell}} \frac{(|S|-1)!(|N|-|S|)!}{(|N|-1)!|N|!} \left[ \tilde{\omega}(S, \sigma) - \tilde{\omega}(S \setminus \{i\}, \sigma) \right] \quad (22)$$

$$= \sum_{\substack{S \subseteq N \\ i \in S}} \sum_{\sigma \in \mathfrak{S}_N} \frac{(|S|-1)!(|N|-|S|)!}{(|N|!)^2} \left[ \tilde{\omega}(S, \sigma) - \tilde{\omega}(S \setminus \{i\}, \sigma) \right] \quad (23)$$

Group the permutations in $\mathfrak{S}_N$ that contain each permutation $\pi \in \mathfrak{S}_S$:

$$= \sum_{\substack{S \subseteq N \\ i \in S}} \sum_{\pi \in \mathfrak{S}_S} \sum_{\sigma \in \Gamma(\mathfrak{S}_N, \pi)} \frac{(|S|-1)!(|N|-|S|)!}{(|N|!)^2} \left[ \tilde{\omega}(S, \sigma) - \tilde{\omega}(S \setminus \{i\}, \sigma) \right] \quad (24)$$

Substitute the characteristic function $\tilde{\omega}$ with $\omega$:

$$= \sum_{\substack{S \subseteq N \\ i \in S}} \sum_{\pi \in \mathfrak{S}_S} \frac{(|S|-1)!(|N|-|S|)!}{|N|!|S|!} \left[ \omega(\pi) - \omega(\pi_{-i}) \right] \quad (25)$$

Note that $\bigcup_{\substack{S \subseteq N \\ i \in S}} \{S\} = \bigcup_{S \subseteq N \setminus \{i\}} \{S \cup \{i\}\}$, therefore:

$$= \sum_{S \subseteq N \setminus \{i\}} \sum_{\pi \in \mathfrak{S}_{S \cup \{i\}}} \frac{(|S \cup \{i\}|-1)!(|N|-|S \cup \{i\}|)!}{|N|!|S \cup \{i\}|!} \left[ \omega(\pi) - \omega(\pi_{-i}) \right] \quad (26)$$

$$= \sum_{S \subseteq N \setminus \{i\}} \sum_{\pi \in \mathfrak{S}_{S \cup \{i\}}} \frac{(|N|-|S|-1)!}{|N|!(|S|+1)} \left[ \omega(\pi) - \omega(\pi_{-i}) \right] \quad (27)$$

Note that we can recover the set of permutations in $\mathfrak{S}_{S \cup \{i\}}$ by taking each permutation $\pi \in \mathfrak{S}_S$ and inserting $i$ at each possible index $k$, i.e.

$$\mathfrak{S}_{S \cup \{i\}} = \bigcup_{\pi \in \mathfrak{S}_S} \bigcup_{k=1}^{|S|+1} \{\pi_{+i,k}\}$$

Therefore, we can rewrite Eq. 27 as follows:

$$= \sum_{S \subseteq N \setminus \{i\}} \sum_{\pi \in \mathfrak{S}_S} \sum_{k=1}^{|S|+1} \frac{(|N|-|S|-1)!}{|N|!(|S|+1)} \left[ \omega(\pi_{+i,k}) - \omega(\pi) \right] \quad (28)$$

This recovers the Sanchez-Bergantiños value [63] for player $i$ in the game $(N, \omega)$. □

## C.3  Proof of Theorem 2

*Proof.* For a given characteristic function $\tilde{\omega}$, we define the following corresponding characteristic function $\omega$, which was introduced in App. C.2:

$$\omega(\pi) = \frac{(|T(\pi)|)!}{|N|!} \sum_{\sigma \in \Gamma(\mathfrak{S}_N, \pi)} \tilde{\omega}\big(T(\pi), \sigma\big) \tag{29}$$

We want to show that the optimal coefficients $\alpha$ for the problem in Eq. 11 using $\mu(s) = \frac{|N|-1}{\binom{|N|}{s}|S|(|N|-s)}$ and characteristic function $\tilde{\omega}$ are equivalent to the SB values under $\omega$.

We first set up the Lagrangian for the optimization problem in Eq. 11, with multiplier $\lambda \in \mathbb{R}$.

$$\mathcal{L}(\alpha, \beta, \lambda) = \sum_{\substack{S \subseteq N \\ S \neq \varnothing, N}} \sum_{\sigma \in \mathfrak{S}_N} \mu(|S|) \left[ \sum_{i \in S} \alpha_i + \sum_{i \in S} \left[\sigma^{-1}(i) - \bar{\ell}\,\right] \beta_i - \left[\tilde{\omega}(S, \sigma) - \tilde{\omega}(\varnothing, \sigma_N^{\mathrm{id}})\right] \right]^2$$

$$-\lambda \left( \sum_{i \in N} \alpha_i - \frac{1}{|N|!} \sum_{\sigma \in \mathfrak{S}_N} \tilde{\omega}(N, \sigma) + \tilde{\omega}(\varnothing, \sigma_N^{\mathrm{id}}) \right) \tag{30}$$

First order conditions with respect to $\lambda$.

$$\frac{\partial \mathcal{L}}{\partial \lambda} = \sum_{i \in N} \alpha_i - \frac{1}{|N|!} \sum_{\sigma \in \mathfrak{S}_N} \tilde{\omega}(N, \sigma) + \tilde{\omega}(\varnothing, \sigma_N^{\mathrm{id}}) = 0 \tag{31}$$

$$\sum_{i \in N} \alpha_i = \frac{1}{|N|!} \sum_{\sigma \in \mathfrak{S}_N} \tilde{\omega}(N, \sigma) - \tilde{\omega}(\varnothing, \sigma_N^{\mathrm{id}}) \tag{32}$$

From Eq. 32, substitute the characteristic function $\tilde{\omega}$ with $\omega$. We change $\sigma$ to $\pi$ for notational consistency.

$$\sum_{i \in N} \alpha_i = \frac{1}{|N|!} \sum_{\pi \in \mathfrak{S}_N} \omega(\pi) - \omega(\pi_\varnothing) \tag{33}$$

First order conditions with respect to $\alpha_i$.

$$\frac{\partial \mathcal{L}}{\partial \alpha_i} = 2 \sum_{\substack{S \subseteq N \\ i \in S \\ S \neq N}} \sum_{\sigma \in \mathfrak{S}_N} \mu(|S|) \left[ \sum_{j \in S} \alpha_j + \sum_{j \in S} \left[\sigma^{-1}(j) - \bar{\ell}\,\right] \beta_j - \left[\tilde{\omega}(S, \sigma) - \tilde{\omega}(\varnothing, \sigma_N^{\mathrm{id}})\right] \right] - \lambda = 0 \tag{34}$$

$$\frac{1}{2}\lambda = \sum_{\substack{S \subseteq N \\ i \in S \\ S \neq N}} \sum_{\sigma \in \mathfrak{S}_N} \mu(|S|) \left[ \sum_{j \in S} \alpha_j + \sum_{j \in S} \left[\sigma^{-1}(j) - \bar{\ell}\,\right] \beta_j - \left[\tilde{\omega}(S, \sigma) - \tilde{\omega}(\varnothing, \sigma_N^{\mathrm{id}})\right] \right] \tag{35}$$

From Eq. 35, separating the components related to $\beta_j$. Note that $\sum_{\sigma \in \mathfrak{S}_N} \left[\sigma^{-1}(j) - \bar{\ell}\,\right] = 0 \ \forall j \in S$.

$$\frac{1}{2}\lambda = \sum_{\substack{S \subseteq N \\ i \in S \\ S \neq N}} \sum_{\sigma \in \mathfrak{S}_N} \mu(|S|) \left[ \sum_{j \in S} \alpha_j - \left[\tilde{\omega}(S, \sigma) - \tilde{\omega}(\varnothing, \sigma_N^{\mathrm{id}})\right] \right]$$

$$+ \sum_{\substack{S \subseteq N \\ i \in S}} \mu(|S|) \sum_{j \in S} \beta_j \underbrace{\sum_{\sigma \in \mathfrak{S}_N} \left[\sigma^{-1}(j) - \bar{\ell}\,\right]}_{=0} \tag{36}$$

Group the permutations in $\mathfrak{S}_N$ that contain each permutation $\pi \in \mathfrak{S}_S$.

$$\frac{1}{2}\lambda = \sum_{\substack{S \subseteq N \\ i \in S \\ S \neq N}} \sum_{\pi \in \mathfrak{S}_S} \sum_{\sigma \in \Gamma(\mathfrak{S}_N, \pi)} \mu(|S|) \left[ \sum_{j \in S} \alpha_j - \left[ \tilde{\omega}(S, \sigma) - \tilde{\omega}(\varnothing, \sigma_N^{\mathrm{id}}) \right] \right] \tag{37}$$

$$\frac{1}{2}\lambda = \sum_{\substack{S \subseteq N \\ i \in S \\ S \neq N}} \sum_{\pi \in \mathfrak{S}_S} \mu(|S|) \left[ \frac{|N|!}{|S|!} \sum_{j \in S} \alpha_j - \sum_{\sigma \in \Gamma(\mathfrak{S}_N, \pi)} \left[ \tilde{\omega}(S, \sigma) - \tilde{\omega}(\varnothing, \sigma_N^{\mathrm{id}}) \right] \right] \tag{38}$$

Substitute the characteristic function $\tilde{\omega}$ with $\omega$.

$$\frac{1}{2}\lambda = \sum_{\substack{S \subseteq N \\ i \in S \\ S \neq N}} \sum_{\pi \in \mathfrak{S}_S} \frac{|N|!}{|S|!} \mu(|S|) \left[ \sum_{j \in S} \alpha_j - \left[ \omega(\pi) - \omega(\pi_\varnothing^{\mathrm{id}}) \right] \right] \tag{39}$$

Sum both sides of Eq. 39 over $i \in N$ and dividing by $|N|$.

$$\frac{1}{|N|} \sum_{i \in N} \frac{1}{2}\lambda = \frac{1}{|N|} \sum_{i \in N} \sum_{\substack{S \subseteq N \\ i \in S \\ S \neq N}} \sum_{\pi \in \mathfrak{S}_S} \frac{|N|!}{|S|!} \mu(|S|) \left[ \sum_{j \in S} \alpha_j - \left[ \omega(\pi) - \omega(\pi_\varnothing^{\mathrm{id}}) \right] \right] \tag{40}$$

$$\frac{1}{2}\lambda = \sum_{i \in N} \sum_{\substack{S \subseteq N \\ i \in S \\ S \neq N}} \sum_{\pi \in \mathfrak{S}_S} \frac{(|N|-1)!}{|S|!} \left[ \underbrace{\mu(|S|) \sum_{j \in S} \alpha_j}_{\textcircled{1}} - \underbrace{\mu(|S|) \left[ \omega(\pi) - \omega(\pi_\varnothing^{\mathrm{id}}) \right]}_{\textcircled{2}} \right] \tag{41}$$

We separate Eq. 41 into two components, $\textcircled{1}$ and $\textcircled{2}$, which we will examine separately.

First, we take $\textcircled{1}$, excluding the outer summation $\sum_{i \in N}$.

$$\textcircled{1} = \sum_{\substack{S \subseteq N \\ i \in S \\ S \neq N}} \sum_{\pi \in \mathfrak{S}_S} \left[ \frac{(|N|-1)!}{|S|!} \mu(|S|) \sum_{j \in S} \alpha_j \right] \tag{42}$$

Change the outer summation to be over subsets of size $s$.

$$= \sum_{s=1}^{|N|-1} \sum_{\substack{S \subseteq N \\ i \in S \\ |S|=s}} \left[ (|N|-1)! \, \mu(|S|) \sum_{j \in S} \alpha_j \right] \tag{43}$$

Separate the terms $i$ and $j \neq i$.

$$= \sum_{s=1}^{|N|-1} \sum_{\substack{S \subseteq N \\ i \in S \\ |S|=s}} (|N|-1)! \, \mu(|S|) \left[ \alpha_i + \sum_{j \in S \setminus \{i\}} \alpha_j \right] \tag{44}$$

$$= \underbrace{\sum_{s=1}^{|N|-1} (|N|-1)! \mu(s) \binom{|N|-1}{s-1} \alpha_i}_{\text{terms involving feature } i} + \underbrace{\sum_{j \in N \setminus \{i\}} \sum_{s=2}^{|N|-1} (|N|-1)! \mu(s) \binom{|N|-2}{s-2} \alpha_j}_{\text{terms involving features } j \neq i} \tag{45}$$

Separating and rearranging the terms involving feature $i$. Note that $\binom{|N|-1}{s-1} = \binom{|N|-2}{s-1} + \binom{|N|-2}{s-2}$.

$$= (|N|-1)! \left[ \sum_{s=1}^{|N|-1} \mu(s) \binom{|N|-2}{s-1} \alpha_i + \sum_{s=2}^{|N|-1} \mu(s) \binom{|N|-2}{s-2} \alpha_i \right. $$
$$\left. + \sum_{j \in N \setminus \{i\}} \sum_{s=2}^{|N|} \mu(s) \binom{|N|-2}{s-2} \alpha_j \right] \quad (46)$$

Combining terms $\binom{|N|-2}{s-2} \alpha_i$ and $\binom{|N|-2}{s-2} \alpha_j$.

$$= (|N|-1)! \left[ \sum_{s=1}^{|N|-1} \mu(s) \binom{|N|-2}{s-1} \alpha_i + \sum_{j \in N} \sum_{s=2}^{|N|-1} \mu(s) \binom{|N|-2}{s-2} \alpha_j \right] \quad (47)$$

Note that $\sum_{j \in N} \alpha_j = \frac{1}{|N|!} \sum_{\pi \in \mathfrak{S}_N} \omega(\pi) - \omega(\pi_\varnothing^{\mathrm{id}})$ from Eq. 33. We can substitute this in the right-most component of Eq. 47.

$$= (|N|-1)! \left[ \sum_{s=1}^{|N|-1} \mu(s) \binom{|N|-2}{s-1} \alpha_i + \sum_{s=2}^{|N|-1} \mu(s) \binom{|N|-2}{s-2} \left[ \frac{1}{|N|!} \sum_{\pi \in \mathfrak{S}_N} \omega(\pi) - \omega(\pi_\varnothing^{\mathrm{id}}) \right] \right] \quad (48)$$

We next sum Eq. 48 over all features $i \in N$ (the outer summation in Eq. 41).

$$\sum_{i \in N} \sum_{\substack{S \subseteq N \\ i \in S \\ S \neq N}} \sum_{\pi \in \mathfrak{S}_S} \left[ \frac{(|N|-1)!}{|S|!} \mu(|S|) \sum_{j \in S} \alpha_j \right] =$$
$$\sum_{i \in N} (|N|-1)! \left[ \sum_{s=1}^{|N|-1} \mu(s) \binom{|N|-2}{s-1} \alpha_i + \sum_{s=2}^{|N|-1} \mu(s) \binom{|N|-2}{s-2} \left[ \frac{1}{|N|!} \sum_{\pi \in \mathfrak{S}_N} \omega(\pi) - \omega(\pi_\varnothing^{\mathrm{id}}) \right] \right] \quad (49)$$

$$= \sum_{s=1}^{|N|-1} (|N|-1)! \mu(s) \binom{|N|-2}{s-1} \sum_{i \in N} \alpha_i + |N|! \sum_{s=2}^{|N|-1} \mu(s) \binom{|N|-2}{s-2} \left[ \frac{1}{|N|!} \sum_{\pi \in \mathfrak{S}_N} \omega(\pi) - \omega(\pi_\varnothing^{\mathrm{id}}) \right] \quad (50)$$

Again, plug in Eq. 33 for $\sum_{i \in N} \alpha_i$.

$$= \sum_{s=1}^{|N|-1} (|N|-1)! \mu(s) \binom{|N|-2}{s-1} \left[ \frac{1}{|N|!} \sum_{\pi \in \mathfrak{S}_N} \omega(\pi) - \omega(\pi_\varnothing^{\mathrm{id}}) \right]$$
$$+ |N|! \sum_{s=2}^{|N|-1} \mu(s) \binom{|N|-2}{s-2} \left[ \frac{1}{|N|!} \sum_{\pi \in \mathfrak{S}_N} \omega(\pi) - \omega(\pi_\varnothing^{\mathrm{id}}) \right] \quad (51)$$

$$= (|N|-1)! \left[ \sum_{s=1}^{|N|-1} \mu(s) \binom{|N|-2}{s-1} + |N| \sum_{s=2}^{|N|-1} \mu(s) \binom{|N|-2}{s-2} \right] \left[ \frac{1}{|N|!} \sum_{\pi \in \mathfrak{S}_N} \omega(\pi) - \omega(\pi_\varnothing^{\mathrm{id}}) \right] \quad (52)$$

$$= (|N|-1)! \left[ \mu(1) + \sum_{s=2}^{|N|-1} s \mu(s) \binom{|N|-1}{s-1} \right] \left[ \frac{1}{|N|!} \sum_{\pi \in \mathfrak{S}_N} \omega(\pi) - \omega(\pi_\varnothing^{\mathrm{id}}) \right] \quad (53)$$

$$= (|N| - 1)! \sum_{s=1}^{|N|-1} s\mu(s) \binom{|N| - 1}{s - 1} \left[ \frac{1}{|N|!} \sum_{\pi \in \mathfrak{S}_N} \omega(\pi) - \omega(\pi_{\varnothing}^{\mathrm{id}}) \right] \tag{54}$$

Next, we take component $\boxed{2}$.

$$\boxed{2} = \sum_{i \in N} \sum_{\substack{S \subseteq N \\ i \in S \\ S \neq N}} \sum_{\pi \in \mathfrak{S}_S} \frac{(|N| - 1)!}{|S|!} \mu(|S|) \left[ \omega(\pi) - \omega(\pi_{\varnothing}^{\mathrm{id}}) \right] \tag{55}$$

$$= \sum_{\substack{S \subseteq N \\ S \neq \varnothing, N}} \sum_{\pi \in \mathfrak{S}_S} \frac{(|N| - 1)!}{(|S| - 1)!} \mu(|S|) \left[ \omega(\pi) - \omega(\pi_{\varnothing}^{\mathrm{id}}) \right] \tag{56}$$

Substituting the results of $\boxed{1}$ $\boxed{2}$ into Eq. 41:

$$\frac{1}{2}\lambda = \underbrace{(|N| - 1)! \sum_{s=1}^{|N|-1} s\mu(s) \binom{|N| - 1}{s - 1} \left[ \frac{1}{|N|!} \sum_{\pi \in \mathfrak{S}_N} \omega(\pi) - \omega(\pi_{\varnothing}^{\mathrm{id}}) \right]}_{\boxed{1}}$$
$$- \underbrace{\sum_{\substack{S \subseteq N \\ S \neq \varnothing, N}} \sum_{\pi \in \mathfrak{S}_S} \frac{(|N| - 1)!}{(|S| - 1)!} \mu(|S|) \left[ \omega(\pi) - \omega(\pi_{\varnothing}^{\mathrm{id}}) \right]}_{\boxed{2}} \tag{57}$$

Set the RHS of Eq. 57 equal to the RHS of Eq. 39

$$\underbrace{\sum_{\substack{S \subseteq N \\ i \in S \\ S \neq N}} \sum_{\pi \in \mathfrak{S}_S} \frac{|N|!}{|S|!} \mu(|S|) \left[ \sum_{j \in S} \alpha_j - \left[ \omega(\pi) - \omega(\pi_{\varnothing}^{\mathrm{id}}) \right] \right]}_{\text{Eq. 39 RHS}}$$
$$= (|N| - 1)! \sum_{s=1}^{|N|-1} s\mu(s) \binom{|N| - 1}{s - 1} \left[ \frac{1}{|N|!} \sum_{\pi \in \mathfrak{S}_N} \omega(\pi) - \omega(\pi_{\varnothing}^{\mathrm{id}}) \right]$$
$$- \sum_{\substack{S \subseteq N \\ S \neq \varnothing, N}} \sum_{\pi \in \mathfrak{S}_S} \frac{(|N| - 1)!}{(|S| - 1)!} \mu(|S|) \left[ \omega(\pi) - \omega(\pi_{\varnothing}^{\mathrm{id}}) \right] \tag{58}$$

Separate the terms in the LHS of Eq. 58 and substitute in Eq. 48.

$$
|N|(|N|-1)! \underbrace{\left[\sum_{s=1}^{|N|-1} \mu(s)\binom{|N|-2}{s-1}\alpha_i + \sum_{s=2}^{|N|-1} \mu(s)\binom{|N|-2}{s-2}\left[\frac{1}{|N|!}\sum_{\pi\in\mathfrak{S}_N}\omega(\pi)-\omega(\pi_\varnothing^{\mathrm{id}})\right]\right]}_{\text{Eq. }48}
$$

$$
- \sum_{\substack{S\subseteq N\\ i\in S\\ S\neq N}}\sum_{\pi\in\mathfrak{S}_S}\frac{|N|!}{|S|!}\mu(|S|)\left[\omega(\pi)-\omega(\pi_\varnothing^{\mathrm{id}})\right]
$$

$$
= (|N|-1)!\sum_{s=1}^{|N|-1} s\mu(s)\binom{|N|-1}{s-1}\left[\frac{1}{|N|!}\sum_{\pi\in\mathfrak{S}_N}\omega(\pi)-\omega(\pi_\varnothing^{\mathrm{id}})\right]
$$

$$
- \sum_{\substack{S\subseteq N\\ S\neq\varnothing,N}}\sum_{\pi\in\mathfrak{S}_S}\frac{(|N|-1)!}{(|S|-1)!}\mu(|S|)\left[\omega(\pi)-\omega(\pi_\varnothing^{\mathrm{id}})\right] \quad (59)
$$

Rearrange and combine terms.

$$
\left[|N|!\sum_{s=1}^{|N|-1}\mu(s)\binom{|N|-2}{s-1}\right]\alpha_i =
$$

$$
\left[\frac{|N|!}{|N|}\sum_{s=1}^{|N|-1} s\mu(s)\binom{|N|-1}{s-1} - |N|!\sum_{s=2}^{|N|-1}\mu(s)\binom{|N|-2}{s-2}\right]\left[\frac{1}{|N|!}\sum_{\pi\in\mathfrak{S}_N}\omega(\pi)-\omega(\pi_\varnothing^{\mathrm{id}})\right]
$$

$$
+ \left[\sum_{\substack{S\subseteq N\\ i\in S\\ S\neq N}}\sum_{\pi\in\mathfrak{S}_S}\frac{|N|!}{|S|!}\mu(|S|)\left[\omega(\pi)-\omega(\pi_\varnothing^{\mathrm{id}})\right] - \sum_{\substack{S\subseteq N\\ S\neq\varnothing,N}}\sum_{\pi\in\mathfrak{S}_S}\frac{(|N|-1)!}{(|S|-1)!}\mu(|S|)\left[\omega(\pi)-\omega(\pi_\varnothing^{\mathrm{id}})\right]\right]
$$

$$
(60)
$$

$$
\left[\sum_{s=1}^{|N|-1}\mu(s)\binom{|N|-2}{s-1}\right]\alpha_i =
$$

$$
\left[\frac{1}{|N|}\sum_{s=1}^{|N|-1} s\mu(s)\binom{|N|-1}{s-1} - \sum_{s=2}^{|N|-1}\mu(s)\binom{|N|-2}{s-2}\right]\left[\frac{1}{|N|!}\sum_{\pi\in\mathfrak{S}_N}\omega(\pi)-\omega(\pi_\varnothing^{\mathrm{id}})\right]
$$

$$
+ \left[\sum_{\substack{S\subseteq N\\ i\in S\\ S\neq N}}\sum_{\pi\in\mathfrak{S}_S}\frac{1}{|S|!}\mu(|S|)\left[\omega(\pi)-\omega(\pi_\varnothing^{\mathrm{id}})\right] - \sum_{\substack{S\subseteq N\\ S\neq\varnothing,N}}\sum_{\pi\in\mathfrak{S}_S}\frac{1}{|N|(|S|-1)!}\mu(|S|)\left[\omega(\pi)-\omega(\pi_\varnothing^{\mathrm{id}})\right]\right]
$$

$$
(61)
$$

Solve for $\alpha_i$.

$$\alpha_i = \frac{1}{|N|}\left[\frac{1}{|N|!}\sum_{\pi\in\mathfrak{S}_N}\omega(\pi)-\omega(\pi^{\text{id}}_\varnothing)\right]$$

$$+\left[\sum_{s=1}^{|N|-1}\mu(s)\binom{|N|-2}{s-1}\right]^{-1}\left[\sum_{\substack{S\subseteq N\\i\in S\\S\neq N}}\sum_{\pi\in\mathfrak{S}_S}\frac{1}{|S|!}\mu(|S|)\left[\omega(\pi)-\omega(\pi^{\text{id}}_\varnothing)\right]\right.$$

$$\left.-\underbrace{\sum_{\substack{S\subseteq N\\S\neq\varnothing,N}}\sum_{\pi\in\mathfrak{S}_S}\frac{1}{|N|(|S|-1)!}\mu(|S|)\left[\omega(\pi)-\omega(\pi^{\text{id}}_\varnothing)\right]}_{\textcircled{3}}\right] \quad (62)$$

In $\textcircled{3}$, we separate the subsets $S\subseteq N$ where $i\in S$ and $i\notin S$, and the subsets $S=N\setminus\{i\},\{i\}$.

$$\textcircled{3}=\sum_{\substack{S\subseteq N\\S\neq\varnothing,N}}\sum_{\pi\in\mathfrak{S}_S}\frac{1}{|N|(|S|-1)!}\mu(|S|)\left[\omega(\pi)-\omega(\pi^{\text{id}}_\varnothing)\right] \quad (63)$$

$$=\sum_{\substack{S\subseteq N\\i\in S\\S\neq\{i\},N}}\left[\underbrace{\sum_{\pi\in\mathfrak{S}_S}\frac{\mu(|S|)}{|N|(|S|-1)!}\left[\omega(\pi)-\omega(\pi^{\text{id}}_\varnothing)\right]}_{\text{Subsets with }i\in S}+\underbrace{\sum_{\pi\in\mathfrak{S}_{S\setminus\{i\}}}\frac{\mu(|S|-1)}{|N|(|S|-2)!}\left[\omega(\pi)-\omega(\pi^{\text{id}}_\varnothing)\right]}_{\text{Subsets with }i\notin S}\right]$$

$$+\underbrace{\sum_{\pi\in\mathfrak{S}_{N\setminus\{i\}}}\frac{\mu(|N|-1)}{|N|(|N|-2)!}\left[\omega(\pi)-\omega(\pi^{\text{id}}_\varnothing)\right]}_{S=N\setminus\{i\}}+\underbrace{\sum_{\pi\in\mathfrak{S}_{\{i\}}}\frac{\mu(1)}{|N|}\left[\omega(\pi)-\omega(\pi^{\text{id}}_\varnothing)\right]}_{S=\{i\}} \quad (64)$$

Substitute $\pi_{-i}$.

$$=\sum_{\substack{S\subseteq N\\i\in S\\S\neq\{i\},N}}\sum_{\pi\in\mathfrak{S}_S}\left[\frac{\mu(|S|)}{|N|(|S|-1)!}\left[\omega(\pi)-\omega(\pi^{\text{id}}_\varnothing)\right]+\frac{\mu(|S|-1)}{|S||N|(|S|-2)!}\left[\omega(\pi_{-i})-\omega(\pi^{\text{id}}_\varnothing)\right]\right]$$

$$+\frac{\mu(|N|-1)}{|N|^2(|N|-2)!}\sum_{\pi\in\mathfrak{S}_N}\left[\omega(\pi_{-i})-\omega(\pi^{\text{id}}_\varnothing)\right]+\frac{\mu(1)}{|N|}\sum_{\pi\in\mathfrak{S}_{\{i\}}}\left[\omega(\pi)-\omega(\pi_{-i})\right] \quad (65)$$

Substituting the results of $\textcircled{3}$ back into Eq. 62.

$$\alpha_i=\frac{1}{|N|}\left[\frac{1}{|N|!}\sum_{\pi\in\mathfrak{S}_N}\omega(\pi)-\omega(\pi^{\text{id}}_\varnothing)\right]+\left[\sum_{s=1}^{|N|-1}\mu(s)\binom{|N|-2}{s-1}\right]^{-1}\times$$

$$\left[\sum_{\substack{S\subseteq N\\i\in S\\S\neq\{i\},N}}\sum_{\pi\in\mathfrak{S}_S}\left[\frac{|N|-|S|}{|N||S|!}\mu(|S|)\left[\omega(\pi)-\omega(\pi^{\text{id}}_\varnothing)\right]-\frac{|S|-1}{|N||S|!}\mu(|S|-1)\left[\omega(\pi_{-i})-\omega(\pi^{\text{id}}_\varnothing)\right]\right]\right.$$

$$\left.-\frac{\mu(|N|-1)}{|N|^2(|N|-2)!}\sum_{\pi\in\mathfrak{S}_N}\left[\omega(\pi_{-i})-\omega(\pi^{\text{id}}_\varnothing)\right]+\frac{(|N|-1)}{|N|}\mu(1)\sum_{\pi\in\mathfrak{S}_{\{i\}}}\left[\omega(\pi)-\omega(\pi_{-i})\right]\right] \quad (66)$$

Eq. 66 provides a general result for a given weight function $\mu : \mathbb{N}^+ \to \mathbb{R}^+$. To complete the proof, we let $\mu(s) = \frac{1}{|N|} \binom{|N|-2}{s-1}^{-1}$.

$$
\begin{aligned}
\alpha_i = \frac{1}{|N|} &\left[ \frac{1}{|N|!} \sum_{\pi \in \mathfrak{S}_N} \omega(\pi) - \omega(\pi_\varnothing^{\mathrm{id}}) \right] \\
&- \frac{1}{|N||N|!} \sum_{\pi \in \mathfrak{S}_N} \left[ \omega(\pi_{-i}) - \omega(\pi_\varnothing^{\mathrm{id}}) \right] + \frac{1}{|N|} \sum_{\pi \in \mathfrak{S}_{\{i\}}} \left[ \omega(\pi) - \omega(\pi_{-i}) \right] \\
&+ \sum_{\substack{S \subseteq N \\ i \in S \\ S \neq \{i\}, N}} \sum_{\pi \in \mathfrak{S}_S} \frac{(|N|-|S|)!(|S|-1)!}{|N|!|S|!} \left[ \omega(\pi) - \omega(\pi_{-i}) \right] \quad (67)
\end{aligned}
$$

$$
\begin{aligned}
= \frac{1}{|N||N|!} \sum_{\pi \in \mathfrak{S}_N} \left[ \omega(\pi) - \omega(\pi_{-i}) \right] &+ \frac{1}{|N|} \sum_{\pi \in \mathfrak{S}_{\{i\}}} \left[ \omega(\pi) - \omega(\pi_{-i}) \right] \\
&+ \sum_{\substack{S \subseteq N \\ i \in S \\ S \neq \{i\}, N}} \sum_{\pi \in \mathfrak{S}_S} \frac{(|N|-|S|)!(|S|-1)!}{|N|!|S|!} \left[ \omega(\pi) - \omega(\pi_{-i}) \right] \quad (68)
\end{aligned}
$$

$$
= \sum_{\substack{S \subseteq N \\ i \in S}} \sum_{\pi \in \mathfrak{S}_S} \frac{(|N|-|S|)!(|S|-1)!}{|N|!|S|!} \left[ \omega(\pi) - \omega(\pi_{-i}) \right] \quad (69)
$$

Note that this result is equivalent to Eq. 25 in the proof of Theorem 1 (App. C.2). We follow the remaining steps in App. C.2 to complete the proof.

$\square$

## C.4 Proof of Corollary 2.1.

We first restate Remark 3 in Sanchez and Bergantiños [63] using our notation.

**Remark.** *Given a game $(N, \omega)$, Nowak and Radzik [53] defined the "averaged game" $(N, \bar{\nu})$ as $\bar{\nu} = \frac{1}{|S|!} \sum_{\sigma \in \mathfrak{S}_S} \omega(S, \sigma) \, \forall S \subseteq N$. Then $\phi(N, \bar{\nu}) = \phi^{SB}(N, \omega)$.*

As we established in Theorem 1, $\bar{\gamma}_i(N, \tilde{\omega}) = \phi^{\mathrm{SB}}(N, \omega)$. Therefore, it follows that Remark 3 in Sanchez and Bergantiños [63] holds for $\bar{\gamma}_i(N, \tilde{\omega})$.

# D Experiment Detail

## D.1 Datasets

**MIMICIII [34].** The Medical Information Mart for Intensive Care (MIMIC-III) dataset contains health records for over 60,000 intensive care unit (ICU) stays at the Beth Israel Deaconess Medical Center from 2001-2012. Each patient's stay is represented as a sequence of codes that include laboratory tests, medications, and drug infusions. We follow the preprocessing in Hur et al. [31], using the provided public repository [3]. To summarize, we create a dictionary using all codes, excluding codes have less than 5 occurrences in the dataset. For codes that contain a numerical component (e.g. drug dosage), we exclude the numerical component to reduce the number of unique codes. We then pass the dictionary to the HuggingFace tokenizer and tokenize each patient's stay using the dictionary, taking only the codes from the first 12 hours of the patient's stay, and limiting the total sequence length to 150 tokens to reduce computational cost. We split the patients into a training set (80%) and test set (20%). After preprocessing, we train a modified BERT model [19] using the

---

[3]https://github.com/hoon9405/DescEmb

Huggingface library [85] with 6 attention heads, 3 hidden layers of width $384$, and a dropout rate of $0.5$.

**EICU [57].** The EICU Colaborative Research Database is a multi-center observational study for over 200,000 ICU stays in the United States from 2014-2015. We use the same preprocessing and model training steps as in MIMICIII but with a different dictionary.

**IMDB [45]**. The Large Movie Review Database (IMDB) contains 50,000 written movie reviews. We use a DistilBert Sequence Classifier [64] to predict positive or negative sentiment. The pretrained model is loaded using the Huggingface library [85]. We then apply XAI methods to infer the importance of each sentence towards the sentiment prediction. We use the NLTK tokenizer [8] to separate each review into sentences. For perturbation-based methods (e.g. KernelSHAP, LIME, OrdSHAP), we pass the individual sentences as "superpixels" for which attributions are calculated. For gradient-based methods (DeepLIFT, IG) we average attributions over each token and sentence.

## D.2 Explainers

**Local Interpretable Model-agnostic Explanations (LIME) [60].** LIME explains individual predictions by training a surrogate linear model locally for each instance to be explained. It perturbs the input data around the sample, weighted by the distance to the original sample, then fits a linear model. The coefficients of the linear model are used as the feature attributions. We use the implementation from the Captum library [37] in our experiments. To ensure fair comparison, we use the same baseline as OrdShap (i.e. the mask token) and disable feature selection, but otherwise use the default parameters.

**KernelSHAP (KS) [42].** KernelSHAP is a model-agnostic method for approximating Shapley values. It builds off the LIME least-squares approach by selecting a weighting that corresponds to Shapley value approximation. We use the official implementation of KernelSHAP from the SHAP library [4]. We use the same baseline as OrdShap (i.e. the mask token) and disable feature selection.

**Leave-One-Covariate-Out (LOCO) [40].** LOCO measures feature importance by comparing a model's prediction with and without a particular feature. For each feature, LOCO removes or randomizes that feature while keeping all others unchanged, then measures the resulting change in prediction. We use the implementation from the Captum library [37] in our experiments.

**DeepLift (DL) [66].** DeepLift is a gradient-based approach that attributes changes in neural network outputs to specific input features. We use the implementation from the Captum library [37] in our experiments. For gradient-based methods, we average the attributions over each token embedding.

**Integrated Gradients (IG) [76].** Integrated Gradients assigns importance to features by calculating gradients along a straight-line path from a baseline to the input. We use the implementation from the Captum library [37] in our experiments. The baseline is set to be the embedding for the mask token. For gradient-based methods, we average the attributions over each token embedding.

## D.3 Metrics

**Insertion / Deletion AUC (iAUC / dAUC) [56]** Insertion/Deletion AUC evaluates explanation quality by measuring changes in model output as features are progressively added or removed. iAUC calculates the area under the curve when pixels or features are added in order of importance, while dAUC measures the curve when they are removed in order of importance. Higher iAUC and lower dAUC indicate better explanations, as they show that the most important features identified have the greatest impact on the model's predictions.

**Inclusion / Exclusion AUC (incAUC / excAUC) [33]** In contrast to iAUC and dAUC, Jethani et al. [33] evaluate the constancy of the masked model outputs to the original model output. This can be measured using *post-hoc accuracy*, which is the accuracy of the masked prediction compared to the original prediction. This post-hoc accuracy metric has also been used in other works [46]. We would expect that masking the most important features from the original sample would lead to a larger drop in accuracy (excAUC). Conversely, including the most important features in a masked sample would lead to a larger increase in accuracy (incAUC). We then calculate the area under the resulting curve for both metrics.

---

[4] https://github.com/shap/shap

Table 4: The associated AUC values for the curves shown in Fig. 4, with standard error in parentheses. The best results are bolded.

| Method | EICU-LOS | EICU-Mortality | MIMICIII-LOS | MIMICIII-Morality | IMDB |
|---|---|---|---|---|---|
| DL | 0.777 (0.015) | 0.724 (0.023) | 0.759 (0.012) | 0.779 (0.011) | 0.869 (0.010) |
| IG | 0.779 (0.014) | 0.746 (0.021) | 0.764 (0.012) | 0.781 (0.011) | 0.874 (0.009) |
| KS | 0.780 (0.014) | 0.752 (0.021) | 0.757 (0.012) | 0.782 (0.011) | 0.870 (0.010) |
| LIME | 0.774 (0.013) | 0.792 (0.018) | 0.758 (0.012) | 0.783 (0.011) | 0.862 (0.010) |
| LOCO | 0.783 (0.014) | 0.775 (0.021) | 0.756 (0.012) | 0.783 (0.011) | 0.870 (0.010) |
| Random | 0.761 (0.012) | 0.781 (0.018) | 0.752 (0.011) | 0.783 (0.011) | 0.870 (0.009) |
| OrdShap-PI | **0.885 (0.005)** | **0.963 (0.007)** | **0.809 (0.010)** | **0.808 (0.010)** | **0.883 (0.008)** |

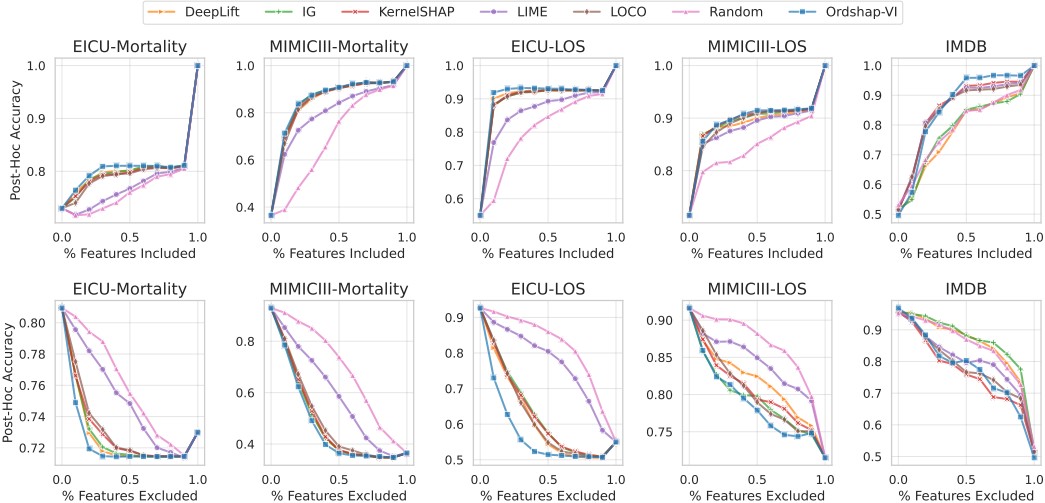

Figure 8: Evaluation of OrdShap-VI using Post-Hoc accuracy when iteratively unmasking features from a baseline sample (Top, higher is better) and masking features from the original sample (Bottom, lower is better). The AUC values for these curves are presented in Table 2.

# E  Additional Results

## E.1  AUC Metrics

**Position Importance.** AUC and standard error results for Fig. 4 are shown in Table 4.

**Value Importance.** The curves associated with Table 2 are shown in Fig. 8 for reference. Additional iAUC and dAUC results are shown in Table 5, with standard error values shown in Table 6.

## E.2  Additional Synthetic Dataset Results

In Fig. 9, we show the results of OrdShap-VI and OrdShap-PI on a synthetic dataset with a linear model.

## E.3  Sensitivity Analysis

In this section, we investigate the sensitivity of OrdShap-PI sample size in the Least-Squares algorithm approximation. We recalculate the position importance metric in §6.1, varying the number of subset samples $K$ and permutation samples $L$, then calculate the resulting standard errors. Results are calculated on 100 explanation samples from each dataset, and shown in Table 7. We observe that the standard error results are relatively stable across different sampling configurations.

Table 5: Insertion AUC and Deletion AUC metrics for evaluating OrdShap-VI. The best results are bolded. Standard error values are provided in Table 6.

| Metric | Model | DL | IG | KS | LIME | LOCO | Random | OrdShap-VI |
|--------|-------|-----|-----|-----|------|------|--------|-----------|
| Insertion AUC ↑ | EICU-LOS | 0.689 | 0.690 | 0.690 | 0.672 | 0.692 | 0.655 | **0.695** |
| | EICU-Mortality | 0.678 | 0.680 | 0.679 | 0.669 | 0.680 | 0.664 | **0.682** |
| | MIMICIII-LOS | 0.692 | 0.695 | 0.694 | 0.687 | 0.692 | 0.661 | **0.696** |
| | MIMICIII-Mortality | 0.697 | 0.701 | 0.700 | 0.677 | 0.697 | 0.633 | **0.702** |
| | IMDB | 0.719 | 0.730 | 0.794 | 0.790 | 0.785 | 0.736 | **0.797** |
| Deletion AUC ↓ | EICU-LOS | 0.611 | 0.611 | 0.610 | 0.643 | 0.608 | 0.661 | **0.597** |
| | EICU-Mortality | 0.646 | 0.645 | 0.646 | 0.655 | 0.647 | 0.661 | **0.643** |
| | MIMICIII-LOS | 0.624 | 0.613 | 0.617 | 0.644 | 0.620 | 0.666 | **0.608** |
| | MIMICIII-Mortality | 0.523 | 0.518 | 0.520 | 0.581 | 0.527 | 0.626 | **0.513** |
| | IMDB | 0.789 | 0.793 | **0.703** | 0.735 | 0.722 | 0.774 | 0.717 |

Table 6: Standard error values for Insertion AUC and Deletion AUC metrics.

| Metric | Model | DL | IG | KS | LIME | LOCO | Random | OrdShap-VI |
|--------|-------|-----|-----|-----|------|------|--------|-----------|
| Insertion AUC ↑ | EICU-LOS | 0.009 | 0.008 | 0.008 | 0.009 | 0.008 | 0.009 | 0.009 |
| | EICU-Mortality | 0.011 | 0.011 | 0.011 | 0.012 | 0.011 | 0.013 | 0.011 |
| | MIMICIII-LOS | 0.008 | 0.008 | 0.008 | 0.008 | 0.008 | 0.009 | 0.008 |
| | MIMICIII-Mortality | 0.008 | 0.008 | 0.008 | 0.009 | 0.008 | 0.011 | 0.008 |
| | IMDB | 0.012 | 0.012 | 0.009 | 0.009 | 0.010 | 0.011 | 0.011 |
| Deletion AUC ↓ | EICU-LOS | 0.011 | 0.011 | 0.011 | 0.009 | 0.011 | 0.009 | 0.012 |
| | EICU-Mortality | 0.017 | 0.017 | 0.017 | 0.014 | 0.016 | 0.013 | 0.018 |
| | MIMICIII-LOS | 0.009 | 0.009 | 0.009 | 0.010 | 0.009 | 0.009 | 0.009 |
| | MIMICIII-Mortality | 0.016 | 0.016 | 0.016 | 0.013 | 0.016 | 0.011 | 0.016 |
| | IMDB | 0.010 | 0.011 | 0.012 | 0.012 | 0.014 | 0.012 | 0.015 |

Figure 9: Attributions for a synthetic dataset and model ($f_\text{linear}$). **(A)** Token values; tokens are assigned Value Importance (VI) and/or Position Importance (PI) with respect to sequence index $i$. **(B)** OrdShap-VI and OrdShap-PI attributions are able to separate the different tokens based on VI and PI effects. **(C)** In contrast, attributions from existing methods cannot distinguish between the different tokens, since VI and PI effects are entangled.

Table 7: Sensitivity analysis of OrdShap-PI sample size in the Least-Squares algorithm approximation. Standard errors are calculated for different combinations of subset samples $K$ and permutation samples $L$ for the position importance metric described in §6.1.

| Permutation Samples $L$ | Subset Samples $K$ | EICU | | MIMICIII | |
|---|---|---|---|---|---|
| | | LOS | Mortality | LOS | Mortality |
| 10 | 10 | 0.0046 | 0.0117 | 0.0100 | 0.0107 |
| | 50 | 0.0046 | 0.0117 | 0.0100 | 0.0107 |
| | 100 | 0.0046 | 0.0117 | 0.0100 | 0.0107 |
| 50 | 10 | 0.0046 | 0.0120 | 0.0099 | 0.0100 |
| | 50 | 0.0046 | 0.0120 | 0.0099 | 0.0100 |
| | 100 | 0.0046 | 0.0120 | 0.0099 | 0.0100 |
| 100 | 10 | 0.0046 | 0.0118 | 0.0103 | 0.0099 |
| | 50 | 0.0046 | 0.0118 | 0.0103 | 0.0099 |
| | 100 | 0.0046 | 0.0118 | 0.0103 | 0.0099 |

