# OpenReview forum: "OrdShap: Feature Position Importance for Sequential Black-Box Models"
_NeurIPS.cc/2025/Conference — NeurIPS 2025 poster_

### Official Review · Reviewer_Sk6y · 2025-06-15

**Clarity:** 3
**Significance:** 4
**Originality:** 4
**Rating:** 5
**Confidence:** 3

**Summary:**

This paper introduces a Shapley-value-based feature attribution method designed to explain the importance of a feature's position within sequential data. To address the high computational cost of Shapley value estimation, the authors propose a least-squares approximation technique. Experimental results using inclusion/exclusion evaluation demonstrate that the proposed method, OrdShap-VI, produces more faithful explanations compared to existing approaches.

**Questions:**

Please see the weaknesses.

**Ethical Concerns:**

["NO or VERY MINOR ethics concerns only"]

**Final Justification:**

The authors' responses have addressed by concerns in the Weaknesses section. I am inclined to Accept.

**Limitations:**

Yes.

**Paper Formatting Concerns:**

No issue is observed.

**Quality:**

3

**Strengths And Weaknesses:**

Strengths:

* The paper addresses a critical limitation in existing feature attribution methods for deep learning models. Most current approaches overlook the role of a feature’s sequential position and focus solely on perturbing feature values. By explicitly modeling positional importance, the proposed method fills a notable gap in the interpretability literature for sequential data.

* The methodology is grounded in a solid theoretical framework, leveraging Shapley values and least-squares approximation to ensure both theoretical soundness and computational feasibility.

Weaknesses:

The primary concern lies in the evaluation strategy of the proposed method.

* Although the method is designed for sequential data, the experiments are limited to a narrow set of domains. While the EHR dataset is sequential in nature, it belongs to a specific subdomain (medical records). The evaluation would be more convincing if the method were tested on standard time-series benchmarks from broader application areas, such as forecasting or regression tasks in finance, weather domains.

* The NLP evaluation is conducted on the IMDB sentiment classification dataset, which is a relatively simple binary classification task. It would be valuable to assess the scalability and robustness of the method in more complex settings—e.g., multi-class classification with thousands of labels, or tasks involving long-context inputs such as those seen in modern large language models (e.g., inputs exceeding 100k tokens). It remains unclear whether the method maintains its sensitivity to positional importance when the input length becomes very large or when the task complexity increases.

---

> ### Author Rebuttal · Authors · 2025-07-31
>
> Thank you for taking the time to review our work. We are encouraged that you appreciate the novelty of our work as well as the theoretical framework. We have provided responses to your concerns and will update the manuscript accordingly. Please feel free to ask any other follow-up questions.
>
> ---
>
> **W1: Experiments are limited to a narrow set of domains**
>
> We thank the reviewer for their feedback. OrdShap is a  general method applicable to any sequential model. We primarily tested on real-world medical record datasets because they represent domains where position effects are particularly consequential and interpretable. In fact, it was through working in this real-world medical domain that inspired us to develop OrdShap.  Note that we also included results on natural language and a synthetic dataset. Based on your suggestion, we agree that demonstrating OrdShap's applicability to other time-series benchmarks would strengthen the evaluation. We will include additional results on a benchmark forecasting task (such as web traffic or climate) in the camera-ready revision.
>
> ---
>
> **W2: Experiments on scalability and robustness of the method**
>
> Thank you for the feedback regarding scalability to (A) multi-class classification with thousands of labels and (B) long-context inputs. Regarding (A), OrdShap follows other Shapley-based methods by calculating separate attributions with respect to each class in a one-vs-all approach. This is because the attributions are directional – their sign indicates importance with respect to one particular class. Therefore, similar to other Shapley methods, OrdShap generally scales linearly with respect to the number of classes to be explained. In particular, if the model outputs for the permuted samples can be stored in memory, then the samples can be reused for each class which further reduces computational cost. In the case of 1000’s of classes, we recommend focusing on the top-k predicted classes or classes of interest. We will edit the manuscript to add more detail based on this discussion.
>
> Regarding (B), the scalability for long-context inputs, we acknowledge that the computational complexity of OrdShap makes it challenging to explain sequences with 1000’s of tokens. For these scenarios, practitioners would likely have to combine tokens into “batches” (e.g. superpixels, text segments, subsequences, etc) and explain batches rather than individual tokens, and/or apply feature selection methods. This cost is inherent to the problem: capturing position-dependent effects requires evaluating the model under different permutations, which is computationally expensive. However, this cost can be justified in applications where the positional effects are important to understand. We also highlight that this work is the first to investigate this notion of positional attributions. We acknowledge this limitation in Section 7, but will add more detail based on this feedback.

---

> > ### Comment · Reviewer_Sk6y · 2025-08-03
> > **Thank you.**
> >
> > Thank you for the clarification. My concerns are addressed. I think this is a quite interesting work and has the potential to large-scale problems, given that the computational complexity is further reduced.

---

### Official Review · Reviewer_dM39 · 2025-07-01

**Clarity:** 3
**Significance:** 3
**Originality:** 3
**Rating:** 5
**Confidence:** 2

**Summary:**

OrdShap, a feature attribution method designed specifically for sequential black-box models, is introduced in this work. It's motivated by observation that existing works like SHAP conflate the importance of a feature's value with the importance of its position within a sequence. A SHAP extension is proposed linking feature-and-position attribution to Sanchez-Bergantinos values, and permutes features in subsets of values. Attributions of value-importance and position-importance are summary metrics from OrdShape. Experiments are done on synthetic data, EHR data, and IMDB to demonstrate the approach.

**Questions:**

* How many permutations were used for experiments in Tab. 2?
* Is there an empirical analysis on the sensitive of the approach to choices of K and L for approximation?
* Is this primarily memory or time bound setup -- what is the maximum sequence length d that's practical?

**Ethical Concerns:**

["NO or VERY MINOR ethics concerns only"]

**Final Justification:**

Feedback from authors seems sufficient. Holding my original rating.

**Limitations:**

yes

**Quality:**

3

**Strengths And Weaknesses:**

* The connection to Sanchez-Bergantinos values to XAI is novel and provides theoretical grounding for the approach.
* The paper overall is well-written, motivated and provides good theoretical and empirical proofpoints.
* Even though acknowledged, the computational cost is absolutely immense: d! * 2^d. LSq-approx is better, but still requires sampling K subsets and L permutations, making it significantly more expensive than KernelSHAP. This also as a result limits the applicability to long sequences (e.g., thousands of doc tokens).
* OrdShap-PI is ultimately just a single linear coefficient. It's likely ultimately an oversimplication and connects weakly to the non-parametric, game-theoretic concept from the theoretical motivation.

---

> ### Author Rebuttal · Authors · 2025-07-31
>
> Thank you for appreciating the novelty and theory of our work. We have provided responses to your clarifying questions below and will update the manuscript accordingly. Please feel free to ask any other follow-up questions.
>
> ---
>
> **W1: Computational Cost**
>
> We acknowledge the reviewer’s concern about computational cost. This cost is inherent to the problem: capturing position-dependent effects requires evaluating the model under different permutations, which is computationally expensive. However, this cost can be justified in applications where the positional effects are important to understand. We also highlight that this work is the first to investigate this notion of positional attributions; there may be methods to further improve efficiency, which we leave for future work.
>
> ---
>
> **W2: OrdShap-PI is simple**
>
> We acknowledge that OrdShap-PI's linear formulation is simple; this is by design for easy interpretation of the OrdShap-PI value for users. Most, if not all, users of attribution methods should be familiar with the meaning of linear coefficients. OrdShap-PI therefore provides an interpretable answer to the question: does this feature matter more when it appears later in the sequence, and by how much more?
>
> ---
>
> **Q1: How many permutations were used in the experiments in Table 2?**
>
> In our experiments we used $L=100$ permutations. During our testing, increasing the number of permutations past 100 showed minimal effect on AUC results.
>
> ---
>
> **Q2: Empirical sensitivity towards choices of K and L?**
>
> Thank you for your suggestion to include a sensitivity analysis on the choice of parameters K (number of subsets) and L (number of permutations). We agree that these results would help guide users in selecting appropriate parameters. We are conducting an empirical sensitivity analysis to show the tradeoff between the number of permutation and subset samples and the variance in OrdShap approximation, which we will include in the camera-ready revision.
>
> ---
>
> **Q3: Memory or Time bound setup?**
>
> The main constraint is typically time rather than memory. Both algorithms require evaluating the black-box model on the permuted and masked samples, which will be the primary bottleneck in practice.
>
> Memory is generally not a practical concern for the following reasons: 1) The black-box model can be queried in batches and only requires a single forward pass per query, 2) the Sampling algorithm can be calculated using batches of samples, and 3) the LS algorithm can be calculated in batches using stochastic gradient descent.
>
> Regarding maximum sequence length: there is no technical limitation due to sequence length, however the number of possible permutations in a sequence increases factorially with $d$, which increases variance when approximating OrdShap with a fixed number of samples. Therefore, for sequences with more than a few hundred features, we recommend first grouping features into interpretable subsets and explaining each group individually, and/or applying a feature selection step to remove irrelevant features. We discuss these strategies in Section 5.2, and will add additional detail in the revised manuscript.

---

### Official Review · Reviewer_fm2z · 2025-07-01

**Clarity:** 4
**Significance:** 3
**Originality:** 3
**Rating:** 4
**Confidence:** 3

**Summary:**

This paper develops OrdShap, a method for attributing predictions of sequential models to both the value and the position of each input feature. Existing attribution approaches typically assume a fixed feature ordering, which makes it difficult to separate the attribution of a feature's value from its position in the sequence. OrdShap addresses this by quantifying how predictions change when the positions of features are permuted.
The method incorporates permutations with the familiar ablations for Shapley values, which the authors connect to Sanchez-Bergantiños values from game theory. It produces two scores per feature: one for its value (OrdShap-VI) and one for its position (OrdShap-PI). The paper also proposes efficient approximation algorithms for computing these scores, using random draws of permutations and feature subsets, and a least squares formulation to calculate attributions for all features at once. Experiments on healthcare, language, and synthetic datasets show that OrdShap can distinguish between value and position effects in model predictions.

**Questions:**

To expand on my comment from "weaknesses":

1) Many models use inputs such as positional encodings to denote a feature's place in the sequence. Why is a simple baseline that adds an input with the position of the feature and runs shap on this extended set of {features, positional inputs} not applicable to the problem? If it is applicable, should the paper be comparing to this?
2) While order of features is an important aspect, isn't it that in most applications, such as the medical example given in the paper, the timing of the feature is important and not just its order? That is, we would have a tuple (t, x) with the time of a medical event/token, and its value, and what matters is not only if a medication was given before/after some token like a lab test, but how much after/before. The paper mentions irregular intervals, says the method can be extended, and points to the appendix. But from the appendix I did not really understand how this extension to such attributions is performed and what properties it has.

Since I am unfamiliar with the literature on methods like TIME and PoSHAP, I would be glad to learn how these works and the current one under review address these questions.

Some smaller comments:
1) Figure 1 is not very easy to understand for a reader that starts reading the paper. It is unclear why the raised predicted probability is a desired outcome and what is it an alternative to. A figure with a clearer story may be helpful for many readers that go through the intro.
2) Section 2 says "While these attributions methods can be applied to sequential models, most assume feature independence". What does "independence" mean here? Is it statistical independence, or some kind of other independence? I guess that KernelSHAP has some independence assumptions, but I was not aware that most shapley-based baselines make statistical independence assumptions.
3) According to theorem 1, OrdShap-VI has the desirable properties that are common in Shap based methods, does OrdShap-PI have any such guarantees as well?

**Ethical Concerns:**

["NO or VERY MINOR ethics concerns only"]

**Final Justification:**

Please see my comment to the rebuttal for the final justification.

**Limitations:**

The authors have appropriately discussed the limitations of their work.

**Paper Formatting Concerns:**

No major concern

**Quality:**

3

**Strengths And Weaknesses:**

Strengths:

The paper is well written and clear. It studies an important problem, and while I am not an expert on attribution methods, it seems like the problem has not been properly addressed before.

Weaknesses:

I think that some motivation for the necessity of the specific permutation-based method would've been beneficial. I will clarify what I mean by that under the "questions" part of the review.
Another small comment is that the evaluation of the order attribution is not entirely clear to me. While the evaluation with Inclusion and Exclusion AUC is pretty intuitive and common for the value attribution, I think the evaluation with permutations at section 6.1 should be better motivated and explained. There is no scalar metric compared between ordshap and and the baselines, and it is also unclear how one should compare it with methods that do not attribute the order at all.

---

> ### Author Rebuttal · Authors · 2025-07-30
>
> Thank you for your thoughtful review and for recognizing the importance of the problem we address and the clarity of our presentation. We appreciate your constructive questions about the motivation for our permutation-based approach and the evaluation methodology. We have provided detailed responses below that we hope will clarify these points. Please feel free to ask any other follow-up questions.
>
> ---
>
> **W1: Clarification of the order attribution evaluation metrics**
>
> The goal of Figure 4 was to evaluate how well OrdShap-PI can capture positional importance. Intuitively, the magnitude of OrdShap-PI answers the question of model sensitivity to permuting the given feature; i.e., if the feature is moved to the beginning or end of the sequence, how strongly does the model output change? The positive/negative directionality of OrdShap-PI indicates whether moving the feature to the end increases the model output (positive OrdShap-PI) or vice versa.
>
> To test this property, we took inspiration from the more standard Inclusion AUC metric for value importance. That is, we first sort OrdShap-PI values based on magnitude. Then, starting from the original sample, we selected an increasing percentage of features (the X-axis in Figure 4) and permute the features to either the beginning or end of the sequence, depending on their OrdShap-PI sign. Similar to Inclusion AUC, we would expect the curve to increase quickly for well-performing attributions.
>
> We follow a similar procedure for competing methods. However, there is ambiguity regarding whether to permute features to the beginning or end of the sequence; we tried both methods and presented the best-performing approach (end of the sequence). We will include an additional plot in the revised manuscript that shows results for the alternate approach (permuting to the beginning of the sequence) to make this more clear.
>
> As we observe in Figure 4, OrdShap-PI performs very well in capturing positional importance. Additionally, as expected, competing attribution methods do not capture positional importance and therefore perform relatively poorly. This experiment also shows that simply applying existing attribution methods is not sufficient if a user wants to understand positional importance.
>
> Regarding scalar metrics, we can convert Figure 4 to an AUC value, which we show in Table 3 in the supplement.
>
> We thank the reviewer for identifying this point of ambiguity. We will add this additional discussion in the revised manuscript.
>
> ---
>
> **Q1: Why not use positional encoding?**
>
> We appreciate the interesting suggestion for applying (e.g.) SHAP to the positional encodings (PE), however this approach is not necessarily trivial. For example, Shapley-based methods generally require defining a baseline value which is used for masking purposes; this is not trivial in the case of PE, since you need to ensure that the masked sequences are valid (e.g. the masked positional encodings cannot be constant “null” values, cannot be repeated in the sequence, must be “valid” PE values, etc.). Additionally, one of the advantages of OrdShap is that it is model-agnostic; it can be applied to any sequential model. A PE-based method would be more limited in the models it can be applied to.
>
> ---
>
> **Q2: Attributions due to feature order vs timing**
>
> Thank you for bringing up this point. We agree that in many applications we are interested in the time that a feature occurs rather than its position. In the main text of our manuscript, we assume that the number of possible permuted positions = the number of features (d) to simplify the notation. However, this can be easily generalized to a time index or any groupings of features, as we describe in App. B.1. Intuitively, the position index we use in the main text can be mapped to any other index, including time. Under this new mapping, for a given feature, OrdShap-PI would give the position importance w.r.t. units of time (rather than position index). OrdShap-VI would give the value importance w.r.t. the mean time of the sequence. We will add more detail to the App. B.1 to clarify this.
>
> Regarding your provided example (paraphrased here), “what matters is not only if a medication was given before some other token, but how much after/before” – we want to clarify that this is a slightly different (though related) problem than what OrdShap solves. The provided example would require *pairwise* positional attributions; i.e. a positional attribution for one feature with respect to another feature. This is similar in concept to feature interaction attributions in XAI literature (e.g. [3]). In contrast, OrdShap generates attributions with respect to a feature’s position/time within the sequence. However, we agree that pairwise positional attributions would be an interesting follow-up work.
>
> ---
>
> **Q3: Figure 1 is not very easy to understand. It is unclear why the raised predicted probability is a desired outcome**
>
> We appreciate the feedback. The purpose of Figure 1 is to help motivate the importance of capturing the effects of feature order in sequential data. Existing attribution methods generally work by quantifying the sensitivity of model output to small changes in the feature *values* for a given sample (see Section 2 for examples), but otherwise assume that the feature order is fixed. However, in Figure 1 we show that prediction models can also be sensitive to changes in feature *order*. In particular, we permute feature order of a given sample while leaving feature values unchanged, and plot the resulting model outputs in the left line plot (in this binary prediction task, model output is the predicted probability of LOS $\geq 3$). We observe that the samples with permuted features often result in different model outputs and predictions (red vs blue lines). This model sensitivity towards feature order cannot be captured by existing methods that assume fixed ordering. We will add this clarification in the revised manuscript.
>
> ---
>
> **Q4: TIME and PoSHAP?**
>
> As mentioned in Section 2, TIME and PoSHAP are attribution methods for sequential models. We can provide a more detailed summary below, which we will add to the revised manuscript.
>
>
> PoSHAP applies KernelSHAP to each sample in a given dataset while also saving the position of each feature in each sample. Then the authors propose to visualize the results as a PxV matrix, where P is the number of possible positions and V is the number of unique features in the dataset, by averaging the KernelSHAP attributions for each position and feature. In contrast with OrdShap, PoSHAP is a global visualization that does not characterize model sensitivity to changing feature value or position for a given sample.
>
> TIME selects a contiguous subsequence of features (referred to as a “time window”) that is important with respect to model prediction. This differs from Ordshap in that it is 1) global, i.e. averaged over the entire dataset rather than w.r.t. individual samples, and is 2) feature selection rather than attribution, i.e. it doesn’t generate an attribution score for each feature. TIME also tests whether the model is sensitive to changes in feature order within each time window; this is again a binary indicator over the entire window and dataset. In contrast, OrdShap generates disentangled, Shapley-based scores for quantifying model’s sensitivity towards a feature’s value and position w.r.t. individual samples.
>
>
> ---
>
> **Q5: Feature Independence for Shapley?**
>
> In this context, we refer to the feature independence assumption in Shapley baselines, sometimes referred to as “off-manifold” [1] or “interventional” [2] Shapley (in contrast with “on-manifold” or “conditional” Shapley). This is the same assumption as used in KernelSHAP, which allows the baseline feature values to be sampled without conditioning on the selected features in each coalition. While this assumption is used in several Shapley methods, we acknowledge that the term “most” may be overstated; we will revise this section and include more detail to clarify this.
>
> ---
>
> **Q6: Shapley Axioms for OrdShap-PI?**
>
> Indeed, while we show that OrdShap-VI satisfies the SB value axioms (Theorem 1) under certain conditions, OrdShap-PI does not inherit these guarantees since its value has a fundamentally different meaning than OrdShap-VI or SB values. Most likely we would need to create a different axiomization for position importance, which could be an interesting direction for future work. We will add this clarification in the “Limitations” section of the paper.
>
> ---
>
> [1] Frye et al, “Shapley Explainability on the Data Manifold”
> [2] Kumar et al, “Problems with Shapley-value-based explanations as feature importance measures”
> [3] Dhamdhere et al, “The Shapley Taylor Interaction Index”

---

> > ### Comment · Reviewer_fm2z · 2025-08-03
> > **Post-rebuttal update**
> >
> > I appreciate the authors' response and plans to introduce changes, like adding explanations about independence assumptions, improved figure, and attributions of time. These address some of the questions in my review.
> >
> > I still think that some kind of baseline which attributes positional embeddings can be pretty useful. The feature for attribution doesn't have to be a positional embedding as used in transformers, but pretty much any feature that assigns an index to each element in the sequence. While there are some choices to make about baseline values, it seems like it'd be useful to have a baseline that makes a somewhat reasonable choice for this, just to be able to compare the proposed method to other related work.
> >
> > Some other points around attributing time, and the feature independence assumption are still not perfectly clear to me, though I trust that the authors will clarify them in the final version in case of acceptance.
> >
> > Overall I am in favor of accepting the paper and will keep my positive score.

---

### Official Review · Reviewer_sQ2K · 2025-07-03

**Clarity:** 4
**Significance:** 3
**Originality:** 4
**Rating:** 5
**Confidence:** 3

**Summary:**

in this paper the authors introduce OrdShap, a novel feature attribution method designed for sequential deep learning models - these often require understanding of both feature values and their positions within input sequences and OrdShap proposes a novel approach to tackle this. Traditional attribution methods assume a fixed feature order, conflating the effects of feature values and their sequence positions. OrdShap addresses this problem by quantifying how model predictions change when feature positions are permuted, thus disentangling the effects of feature values and positions which ultimately shows superior performance. The method is grounded in game theory, specifically relating to Sanchez-Bergantinos values (which the authors do a good job of introducing and motivating) and provides a structured representation of feature importance based on position. Empirical evaluations on health, natural language, and synthetic datasets demonstrate that the proposed method, OrdShap, is effective in capturing both feature value and position attributions, offering deeper insights into model behavior. The paper also proposes efficient algorithms to approximate OrdShap, addressing computational challenges inherent in Shapley-based methods. The experimental evaluations show benefits in terms of both interpretability as well as superior performance to several benchmarks.

**Questions:**

Please see strengths/weaknesses.

**Ethical Concerns:**

["NO or VERY MINOR ethics concerns only"]

**Final Justification:**

The authors gave clear examples and added more experiments which answered my questions.

**Limitations:**

Yes

**Paper Formatting Concerns:**

No concerns

**Quality:**

3

**Strengths And Weaknesses:**

The paper is very well written and the motivation is clear. The authors give a nice introduction and the technical preliminaries section is well written. One thing I really like about this work is that authors are very pedagogical about their approach and explanations. There is consistently a rigorous theoretical explanation and alongside it a intuitive explanation or a motivating example showcasing what it means. The sanchez and Bergantinos values are also well motivated and if it is the case that they have not been investigated in XAI literature (in which I am not well versed but could not find many works online combining the two), I think the idea is neat and clear.

Regarding the introductory example of predicting LOS, could the authors give an example of the "medical tokens" and how does it look in practice, maybe in appendix in a small section? Although I really appreciate the figure 2 and it made the example much clearer, I think this would be the cherry on top that would make the example perfectly clear. For many of us who have been working with LLMs the word token and its meaning is now a bit contrived, I think this would possibly clear any remaining confusion that the reader might have.

I would once again like to emphasize my satisfaction with the authors' ability to straightforwardedly explain the concepts, as I found the Table 1 and the example given with Bag, Gloves and Hats very intuitive and nice to motivate the need for using OrdShap.

Although the method is clear, the question of computational expense naturally arises but the authors did a good job of both presenting it as the technical limitations of the paper and did a good job in producing sampling and least-squares approximations. I am not too sure if the least-squares approximation can give any downsides and what are its weaknesses, and I would appreciate if the authors could comment a bit more on this but in general I was satisfied with both approaches that authors proposed (5.1 and 5.2). Regarding experiments, I was convinced with the benefit of using OrdShap and was happy that authors gave execution time results as well as presented their details so detailedly. However, looking at Table 7 in the supplementary material I was a bit surprised with how much slower the OrdShap methods are and I think that the authors should have highlighted this in the main paper as it is extremely slower. Although it is mentioned in the limitations and conclusion section, the authors a bit misleadingly mention that they have mitigated this limitation through sampling and approximation. Also, I think that the significance results should be reported in the main paper and not the supplementary material - especially as (unless I am mistaken), most of the results for OrdShap-VI in the e.g., Inclusion AUC are not statistically significantly different to the performance of second and third best performing method (e.g., IG and KS).

Overall I think it is a well written paper that contributes to XAI with clear motivation and applications but there are few weaknesses that I think the authors should address and comment upon. I think that the authors open a nice alleyway for algorithms that can disentangle value and position importance of each feature in a sequence and introducing the SB values is definitely something that can be important/useful for XAI, but after examining the appendix and the statistical (in)significance of some results that come at an expense of much longer computation and execution times, I was reluctant to give a larger score.

---

> ### Author Rebuttal · Authors · 2025-07-30
>
> Thank you for your thoughtful feedback and for appreciating the clarity of our presentation and the novelty of introducing Sanchez-Bergantinos values to XAI. We are glad that you found our examples helpful. We have provided responses to your concerns below, which we hope will help provide additional clarification. Please feel free to ask any other follow-up questions.
>
> ---
>
> **Q1: Examples of Medical Tokens**
>
> We appreciate the suggestion of including additional explanation and examples of medical tokens. In our experiments and examples we use electronic health record (EHR) data, which consist of clinical events that occur during a patient’s interactions with hospitals or clinics. Clinical events can include medical diagnosis (such as ICD9 codes), prescribed medications, medical procedures, or laboratory tests. Each sample in our EHR datasets contains the history for an individual patient, which consists of lists of these clinical events over time. Recent works (e.g. [1,2]) have explored tokenizing these clinical events for use with Transformers in various prediction tasks. While there are a variety of approaches, most convert each unique clinical event to a separate token, which we refer to as medical tokens in the manuscript. Therefore, the processed EHR datasets in our paper are tokenized samples, where each sample represents a separate patient, and each token represents a clinical event.
>
> Below we provide some examples of medical tokens. The MIMICIII and EICU datasets include medications, laboratory tests, and infusions.
>
> Medications:
> * Sodium Chloride 0.9% IV
> * Insulin Lispro (Human) 100 unit/mL
> * Dextrose 50% IV Solution
> * Hydrocortisone Sodium Succinate PF 100 mg
> * Magnesium Sulfate 50% Injection Solution
>
> Laboratory Tests:
> * total bilirubin
> * platelets x 1000
> * WBC x 1000 [White blood cell count]
> * MPV [Mean Platelet Volume]
> * Glucose
>
> Infusions:
> * Normal Saline 20K
> * Epinephrine
> * Sodium Bicarbonate mL/hr
> * Propofol
> * Isoproterenol
>
>
> We will add this additional clarification to the revised manuscript.
>
>
> ---
>
> **Q2: Downsides of the Least Squares Algorithm**
>
> Thank you for raising this point. The main limitation of the Least Squares (LS) Algorithm compared to the Sampling algorithm is that it only approximates the summary metrics OrdShap-PI and OrdShap-VI, and not the full OrdShap values $\gamma_{i,\ell}$ (Eq. 8). Users who need to calculate the full OrdShap values, such as for visualization (as in Figure 3B) or for analyzing nonlinear position effects, would need to use the Sampling algorithm. Since many users primarily need the summary metrics for interpretation, this tradeoff is often worth it.
>
> A second, minor limitation is that the LS algorithm always solves a weighted least squares problem with $K \times L$ samples and computes attributions for all features simultaneously. For very small/fast models, or when only a few features need attribution, the Sampling algorithm could theoretically be more efficient. However, in most applications the LS algorithm is generally faster than the Sampling algorithm.
>
> We will add this discussion to the revised manuscript to improve clarity.
>
> ---
>
> **Q3: Time Complexity Concerns**
>
> We appreciate the feedback regarding time complexity. We placed the empirical time complexity results (Table 7) in the supplement due to space constraints. We are happy to move these results to the main text in the camera-ready revision, which allows for an additional page of content.
>
> Our claims of efficiency improvement are with respect to the naive implementation of Eq. 8, which is generally infeasible for any non-toy dataset. Our proposed Sampling and LS algorithms make OrdShap values computationally feasible in practice. We also highlight that this work is the first to investigate this notion of position-dependent attributions, the other methods presented in Table 7 are not able to capture this type of attribution.
> Ultimately, capturing position-dependent effects requires evaluating the model under different permutations, which is computationally expensive. This computational cost can be justified in applications where the positional effects are important to understand.
>
> We will make this time complexity limitation more clear in the revised manuscript to avoid misunderstanding.
>
>
>
> ---
>
> **Q4: Statistical Significance of Results**
>
> Similar to the previous response, we would be happy to move the standard error results to the main text, given the extra space for the camera-ready revision. We acknowledge that some of the OrdShap-VI AUC scores are not statistically significant. We highlight that OrdShap-VI still performs competitively compared with the top-performing traditional attribution methods. While these other methods still perform relatively well in generating attributions for *value importance*, they are unable to disentangle and provide the effects of *position importance*, which we are able to do with OrdShap-PI.

---

> > ### Comment · Reviewer_sQ2K · 2025-08-03
> >
> > Thank you for your reply and I look forward to the revised version of the paper.
> >
> > I am also very pleased by the ability to clearly state and give examples as the authors do in the paper as well as in their response. I am happy to increase my score to accept. Also, thank you for being honest about the statistical significance of your work - your openness is much appreciated.

---

### Decision · Program_Chairs · 2025-09-17

**Decision:**

Accept (poster)

**Comment:**

This paper proposes OrdShap, a Shapley-value-based attribution method for sequential models that disentangles the influence of feature values from their positions on model predictions. The proposed approach builds on Sanchez-Bergantiños values from game theory, and uses efficient sampling and least-squares approximations to address the computational cost of exact computation. Experiments on synthetic, healthcare, and NLP datasets validate the approach.

Reviewers praised the paper’s clarity, pedagogical examples, and the novelty of connecting positional attribution to Sanchez-Bergantiños values. Initial concerns regarding computational cost, evaluation scope, and statistical significance of reported results were satisfactorily addressed in the rebuttal. Overall, the contribution is well-motivated, technically sound, and has potential to make a significant impact in explainable AI for sequential models. The authors are encouraged to revise their final version according to the rebuttal, including clarifying figures, attributing time, and the feature independence assumption.